



# Investigating spatiotemporal patterns of snowline altitude at the end of melting season in High Mountain Asia, using cloud-free MODIS snow cover product, 2001-2016

**Zhiguang Tang [1],\*, Xiaoru Wang [1], Jian Wang [2], Xin Wang [1], Junfeng Wei [1]**

[1] National-Local Joint Engineering Laboratory of Geo-spatial Information Technology, Hunan University of Science and Technology, Xiangtan 411201, China

[2] Key Laboratory of Remote Sensing of Gansu Province, Northwest Institute of Eco-Environment and Resources, Chinese Academy of Sciences, Lanzhou 730000, China

**\*Correspondence to**: Zhiguang Tang ( tangzhg11@hnust.edu.cn)

**Abstract:** The snowline altitude at the end of melting season (SLA-EMS) can be used as an indicator of the equilibrium line altitude (ELA) and therefore for the annual mass balance of glaciers in certain conditions. High Mountain Asia (HMA) hosts the largest glacier and perennial snow cover concentration outside the polar regions, but the spatiotemporal pattern of SLA-EMS under climate change is poorly understood in there. Here, we develop a method for estimating SLA-EMS over large-scale area by using the cloud-removed MODIS fractional snow cover data, and investigate the spatiotemporal characteristics and trends of SLA-EMS during 2001-2016 over the HMA. The possible linkage between the SLA-EMS and temperature and precipitation changes over the HMA is also investigated. The results are as follows: (1) There are good linear regression relationships (R = -0.66) between the extracted grid (30km) SLA-EMS and glaciers annual mass balance over the HMA. (2) Generally, the SLA-EMS in the HMA decreases with increase of latitude. And due to the mass elevation effect, it decreases from the high altitude region of Himalayas and inner Tibet to surrounding low mountainous area. (3) The SLA-EMS of HMA generally shows a rising trend in the recent years (2001-2016). In total, 75.3% (24.2% with a significant increase) and 16.1% (less than 1% with a significant decrease) of the study area show increasing and decreasing trends in SLA-EMS, respectively. The SLA-EMS significant increases in Tien Shan, Inner Tibet, south and east Tibet, east Himalaya and Hengduan Shan. (4) Temperature (especially the summer temperature) trends to be the dominant climatic factor affecting the variations of SLA-EMS over the HMA. Under the background of the generally losing glaciers mass in HMA, if the SLA-EMS continues to rise as a result of global warming, it will accelerate the negative mass balances of the glaciers. This study is an important step towards reconstruction the time series of glacier annual mass balance using SLA-EMS datasets at the scale of HMA to better document the relationships between climate and glaciers.

**Keywords:** snowline altitude; spatiotemporal patterns; High Mountain Asia; MODIS

## 1. Introduction

Snow and glaciers play an important role in the global energy and water cycles because of their high albedo and water storage properties, and can indicate the changes in global climate. Studying the variations of glaciers and





snow is of significant importance in monitoring and maintaining water management for ecosystem processes
and irrigation practices, because of there are more than one sixth of the global population relies on water from
mountainous melt water (Barnett et al., 2005).
Snowline altitude and its inter- and intra-annual variability are key characteristics indicating temporal
changes in snow cover and duration of snow melt (Krajčí et al., 2014). The concept of snowline estimation
varies with the applications. In hydrological applications, the snowline is identified as the boundary separating
snow-covered areas from snow-free areas (Kaur et al., 2010; Parker, 1997; Seidel et al., 1997), for estimation of
snow covered area and its temporal evolution, which is hence used as an input for hydrological modeling
(Holzer et al., 1995; Martinec et al., 2008), or for validation of snow model simulations (Turpin et al., 1997;
Zappa, 2008). And, the snowline altitude estimates have also been applied as an alternative method for cloud
removal in satellite snow cover products (Gafurov and Bárdossy, 2009; Parajka et al., 2010). In glacier and
climate studies, the snowline defines the lowest altitude of the perennial snow cover (Flint, 1971), equivalent to
the lower boundary of the snow covered area at the end of melting season (also known as snowline at end of
melting season (Pandey et al., 2013)). The snowline altitude at the end of melting season (SLA-EMS)
approximates the equilibrium line altitude (ELA), it can serves as a good proxy for ELA and therefore for the
mass balance of glaciers (McFadden et al., 2011; Pandey et al., 2013; Rabatel et al., 2005, 2012; Tawde et al.,
2016). Numerous studies (Braithwaite, 1984; Rabatel et al., 2005, 2008, 2012; WGMS, 1991-2013; Xie et al.,
1996) have shown that glacier annual mass balance is highly correlated with the ELA and SLA-EMS, and it
enables reconstruction of annual mass balance time series. The climate sensitivity of SLA-EMS has been
generally emphasized as a supplement to current climate change indicator systems. A study of the
spatial-temporal variations of the SLA-EMS can help in assessing the hydrologic cycle balance as well as to
understand the regional and global cryosphere and climate changes.
High Mountain Asia (HMA) hosts the largest perennial snow and glacier concentration outside the
Antarctic and Arctic regions. As one of the most sensitive and prominent areas responding to global climate
changes, the temperature of the HMA has increased more than twice the rate of global warming in the past
decades (Hu et al., 2013; Qiu, 2008); and the snow cover and glaciers in HMA are in a state of rapid change
(Brun et al., 2017; Rittger et al., 2016; Yao et al., 2012). These have led to changes in mountainous hydrological
processes and water resources, and increased the HMA's runoff due to the accelerated glacier/snowmelt (Chen
et al., 2017; Duethmann et al., 2015; Lutz et al., 2014). Meltwater from snow and glaciers in HMA provides a
major source of water for approximately 1.4 billion of inhabitants in the downstream low lying plains
(Immerzeel et al., 2010). Changes in the SLA-EMS with the increase or reduction of perennial snow- and
ice-covered area have a profound effect on stream water availability in the basins (Lei et al., 2012). For the
future livelihood of these people, it is therefore very important to estimate the SLA-EMS in HMA, understand
how the SLA-EMS will respond to climate changes.
The traditional strategy for SLA-EMS estimating is direct ground-based observations using field methods,
and extrapolating them to the rest of the mountain ranges. In HMA, this strategy suffers from the scarcity of
local observations in space and time and the consequent need to extrapolate to vast unsampled areas. This is
problematic, given that the pattern of glacier change in HMA is now known to be strongly heterogeneous (Brun
et al., 2017). Satellite remote sensing offers the opportunity to extract and evaluate snowline in the inaccessible
areas with rugged terrain and hostile climate. Most previous studies (McFadden et al., 2011; Pandey et al., 2013;
Rabatel et al., 2005, 2012; Tawde et al., 2016; Zhang and Kang, 2017) of SLA-EMS have focused on local areas



and using visual interpretation of Landsat MSS/TM /ETM+ and SPOT images observed in the end of summer
in which the snow cover extent is at its minimum value. However, they are difficult to assess SLA-EMS
changes in a continuous time and space for a large-scale area, due to the 16-day or longer revisit period and
relatively small swath width of Landsat and SPOT.

The most recent and advanced remote sensing snow cover product is produced by MODIS flown on the

Earth Observing System (EOS) Terra and Aqua platforms. The MODIS snow cover products (Riggs et al., 2006)
have been widely used to depict the spatiotemporal patterns of the seasonal or transient snowline altitude in
mountainous areas (Krajčí et al., 2014; Krajčí et al., 2016; Parajka et al., 2010; Spiess et al., 2016; Tang et al.,
2014; Verbyla et al., 2017), although they were rarely used to evaluate the SLA-EMS. Evaluation studies have
suggested a high accuracy of MODIS snow cover products under clear skies, when comparing with the in-situ
observations and other higher resolution satellite data at both regional and global scales (Hall and Riggs, 2007;
Klein et al., 2003; Tang et al., 2013). However, the extensive cloud obscuration in MODIS snow cover products
greatly limits their applications (Hall and Riggs, 2007; Riggs et al., 2006). Due to the MODIS fractional snow
cover (FSC) maps more accurately represent the gradual changes of snow cover in each pixel than the binary
snow cover maps (Riggs et al., 2006; Salomonson and Appel, 2004), the use of the FSC data could be better for
the removal of the cloud cover by temporal filtering. Consequently, we have developed a cubic spline
interpolation cloud removal method to eliminate the cloud covered pixels from the MODIS FSC products (Tang
et al., 2013), and the cloud removal method was well applied in Tibetan Plateau and Tianshan Mountains (Tang
et al., 2013, 2014, 2017). The advantage of the cloud removed MODIS FSC products is that the FSC changing
curve (the characteristic of gradual changes) for each pixel are considered rather than the simple substitution by
multiday combination for the binary snow cover products (Wang and Xie, 2009; Xie et al., 2009; Zhang et al.,
2016b), and all of the cloud pixels and other missing or abnormal pixels are removed with a high
snow-classification accuracy, thus providing considerable application value for snow cover monitoring.

The objectives of this paper are to: (1) propose a method for estimation of SLA-EMS over a large-scale

area, based on the cloud-removed daily MODIS FSC data; (2) give spatially detailed estimates of the changes of
SLA-EMS in HMA during 2001-2016 on a grid-by-grid basis. We also examine the SLA-EMS changes from
different subregions of the HMA, and the possible cause for the SLA-EMS changes from the perspective of
climate factors (temperature and precipitation). The topic is important as it will help to improve our
understanding of the climate-cryosphere relationship at the scale of a whole HMA. This study will also be an
important step towards reconstruction the time series of glacier annual mass balance using SLA-EMS datasets
at the scale of HMA to better document the relationships between climate and glaciers.
**2. Study area and data**
**2.1. Study area**
The HMA is the largest and highest mountain region on earth, stretching across Central Asia between 65°-
105°E and 25°- 51°N (Figure 1). It covers an area over 5 million km2, with the average elevation over 4000 m,
spanning regions from the Hindu Kush and Pamir in the west to the Hengduan Mountains in the east, from
Himalayas in the south to the Altay and Sayan in the north. Based on mountain ranges, HMA was subdivided
into 16 subregions (Figure 1).

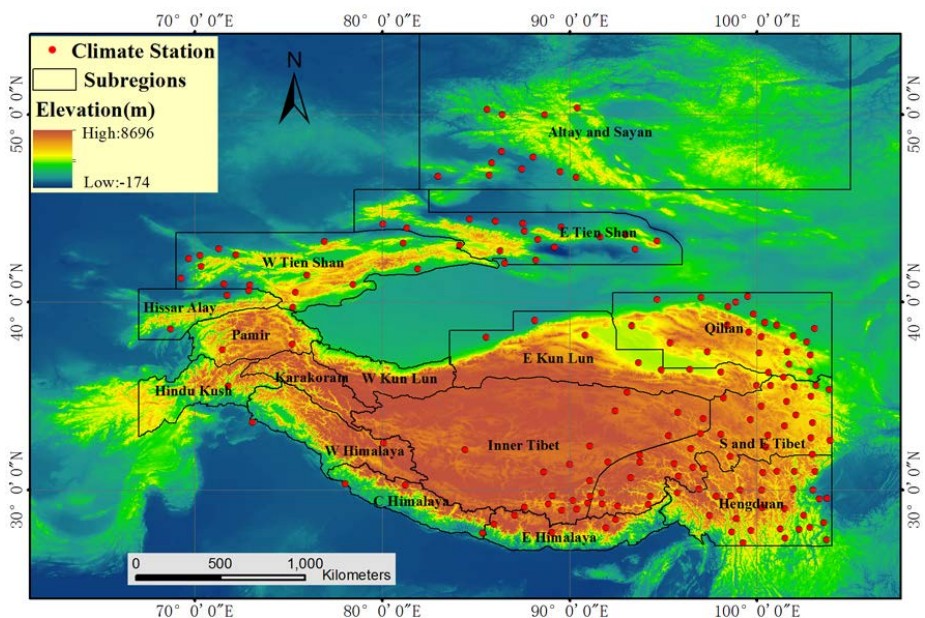

**Figure 1.** Location and the extent of the study area and subregions. Red dots denote meteorological stations. In names of subregions, these individual letters 'W', 'C', 'E' and 'S' mean west, central, east and south, respectively.

## 2.2. Data

### 2.2.1. MODIS Fractional Snow Cover (FSC) Data

In this study, the MOD10A1 data for 2001–2016 from the National Snow and Ice Data Center (NSIDC) (https://nsidc.org/) are employed to investigate the SLA-EMS variations in the HMA. MOD10A1 data are daily snow cover products, which include both binary snow cover and fractional snow cover (FSC) products (with a resolution of 463.3 m), gridded in sinusoidal projection (Riggs et al., 2006). The MODIS FSC mapping algorithm is developed by Salomonson and Appel (Salomonson and Appel, 2004), which is based on a statistical-linear relationship developed between the normalized-difference snow index (NDSI) from MODIS and the true subpixel fraction of snow cover as determined using Landsat scenes. Evaluation studies have proved a high accuracy (with a mean absolute error less than 0.1) of the MODIS FSC data (Hall and Riggs, 2007; Salomonson and Appel, 2004; Tang et al., 2013) . Using the MODIS Reprojection Tool (MRT) (Dwyer and Schmidt, 2006), the MOD10A1 FSC data are mosaicked and resampled from the original 463.3 m pixel size to 500 m, and georeferenced into a UTM projection with a datum of WGS84. The final mosaicked images are converted to GeoTIFF file format.

### 2.2.2. Meteorological Observation Data

Daily temperature and precipitation from 172 meteorological stations (Figure 1) in the HMA for 2001–2016 are collected from China Meteorological Administration (http://data.cma.cn) and NOAA's National Centers for Environmental Information (NCEI) (https://www.ncdc.noaa.gov/), formerly known as National Climatic Data

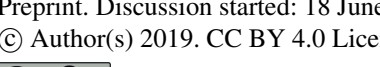



Center. The temperature and precipitation data are used to survey the linkages between SLA-EMS changes and
climate variations during the study periods.

### 2.2.3. Glacier Annual Mass Balance Observation Data

The Fluctuations of Glaciers Database (2017)((WGMS), 2017) (DOI: 10.5904/wgms-fog-2017-10) is collected
from World Glacier Monitoring Service (WGMS, https://wgms.ch/data_databaseversions/). It is an
internationally collected, standardized dataset on changes in glaciers (length, area, volume and mass), based on
in-situ and remotely sensed observations, as well as on reconstructions. In this study, 12 observed glaciers
(Figure 2) with the annual mass balance observations over 6 years (and 8 of them are observed over 10 years)
are selected as calibration data for the SLA-EMS estimation.

### 2.2.4. Other Data

The digital elevation model (DEM) data at the spatial resolution of 90 m from the Shuttle Radar Topography
Mission (SRTM) are used to derive the altitude values of SLA-EMS. They are available at
http://srtm.csi.cgiar.org/. In order to match the MODIS images, the SRTM DEM are resampled from the
original 90-m pixel size to 500 m. The glacier inventory data (Figure 2) used in this study includes the Second
Glacier Inventory Dataset of China (http://westdc.westgis.ac.cn/data) and the Randolph Glacier Inventory 5.0
(RGI 5.0, http://www.glims.org/RGI/randolph50.html); the RGI 5.0 is used for the study areas of outside China.
Besides, several Landsat TM/ETM+/OLI images of the five selected Landsat scenes (Figure 2) are also used as
reference data for the SLA-EMS estimation.
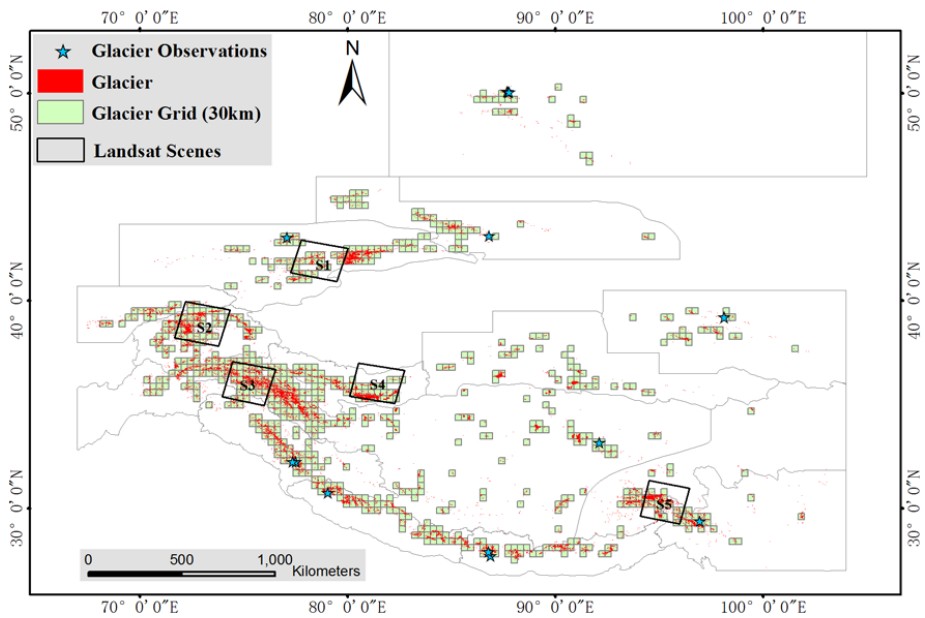

**Figure 2.** Spatial distribution of glacier grids, glacier observations and Landsat scenes. The annual mass
balance observations of 12 observed glaciers (listed in Table 1) are collected from the Fluctuations of Glaciers
Database (https://wgms.ch/data_databaseversions/). Glacial areas in the HMA are divided into 744 glacial





grids with the cell size of 30km. The five black boxes (S1-S5) outline the area covered by Landsat images (in
Table 2).

## 3. Methods

### 3.1. Cloud Removal from MODIS FSC Data, and SCD Calculation

Following the cloud removal method for MODIS FSC products developed by Tang et al. (Tang et al., 2013), the
daily cloud-removed FSC data in the study area were produced from 2000 to 2016. The cloud removal
algorithm is based on the cubic spline interpolation algorithm (temporal filtering). Details on the cubic spline
interpolation cloud removal method and the relevant accuracy evaluation strategies can be found in the works of
Tang et al. (Tang et al., 2013). From the applications of the cloud removal method in the areas of HMA (Tibetan
Plateau and Tianshan Mountains) (Tang et al., 2014; Tang et al., 2013; Tang et al., 2017), the cloud removal
method was efficient in retrieving the FSC information of these cloud covered pixels in these areas, with the
overall mean absolute error less than 0.1; and there was a high consistency between MODIS-derived
snow-covered days (SCD) and the in-situ observed SCD, the mean consistency over 85%, and the mean
absolute error is less than 4.2 days (Tang et al., 2014; Tang et al., 2013; Tang et al., 2017). The higher
consistency between MODIS-derived SCD and in-situ SCD indicates that the cloud-removed MODIS FSC data
have a high accuracy to monitor the snowline in the HMA.
The SCD represents the overall snow cover conditions for a region in a year. Therefore, the maps of the
spatial distribution of SCD have potential significance for SLA-EMS determination. In this study, the SCD
images are calculated using all cloud-free MODIS FSC images for a given year. The calculation equation is
shown as:
$$\text{SCD} = \sum_{i=1}^{N} Ceil(D_i \geq 50) \tag{1}$$

where N is the total number of days (images) within a year and $D_i$ is the snow cover fraction (%) in a
pixel($0 \leq D_i \leq 100$). Ceil ($D_i \geq 50$) counts the numbers of $D_i \geq 50$. For instance, if the pixel value on the image is 60
(i.e., 60% snow cover), the SCD adds 1. If the pixel value on the image is 10 (i.e., 10% snow cover), the SCD
adds 0 and is unchanged. Figure 3 shows the distribution of the MODIS-derived SCD in the HMA.

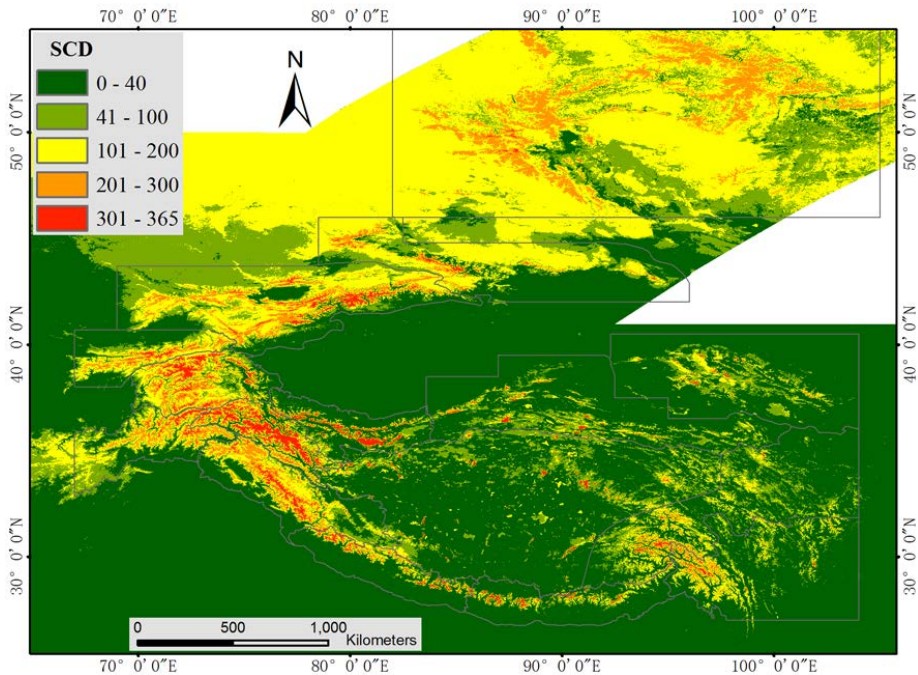

**Figure 3.** Distribution of average snow-covered days (SCD) from 2001 to 2016 in the High Mountain Asia (HMA).

### 3.2. Methodology of SLA-EMS monitoring

For snow cover, the end time of melting season varies with the places and years, so that perennial snow cover (i.e., the snow cover at the end of melting season) in a range of HMA is difficult to identify in terms of the snow cover on a specific given date. Theoretically, the SLA-EMS can be determined by the MODIS derived snow covered days (SCD), that is, the boundary altitude of perennial snow cover (where the SCD≥365d). However, the snow area with MODIS SCD≥365d is fail to really indicates the perennial snow area, due to the affection of the annual cumulated errors in MODIS snow mapping algorithm and cloud removal method. For instance, our previous studies (Tang et al., 2013; Tang et al., 2017) in Tibetan plateau and Tianshan Mountains have found that the snow areas with MODIS SCD≥365d are far less than the glacier areas, although the clouds in MODIS snow product are effectively eliminated. In addition, how to estimate the boundary altitude value of the perennial snow cover is another problem need to be solved. Therefore, the designed methods of monitoring the SLA-EMS from cloud-removed MODIS FSC products in this study involve (1) dividing glacier grids, (2) calibration of MODIS SCD threshold for perennial snow cover estimation, (3) determination the boundary altitude value of the perennial snow cover.

### 3.2.1. Dividing Glacier Grids

In order to make detailed and comprehensive assessments of the SLA-EMS, glacial areas in the HMA are divided into 744 grids (named glacier grids) with the cell size of 30km (Figure 2). For each identified glacier grid, the area of glacier cover is ensured to be more than 25km2 based on overlay analysis with the glacier





inventory data. Thus, the SLA-EMS in the HMA can be investigated thoroughly on a grid-by-grid and
year-by-year basis.

**3.2.2. Calibration of MODIS SCD Threshold for Estimating Perennial Snow Cover**

Since SLA-EMS is able to serves as a good proxy for ELA and thus for the mass balance of glaciers, it is
reasonable to calibration of MODIS SCD threshold value by comparing the correlations between the SLA-EMS
and annual mass balance of glaciers. In this method, the annual mass balance observations of the 12 measured
glaciers (Figure 2) are used as reference data for SCD threshold calibration, and yearly SLA-EMS values of the
corresponding glacier grids are extracted for any given SCD values (setting the SCD threshold changing from
280 to 365d, with 1d as the step size). Here, the method of SLA-EMS values extraction is shown in section 3.2.3.
Figure 4 shows the average correlation coefficients between annual mass balance of the 12 glaciers and the
SLA-EMS changing with the MODIS SCD threshold. As the MODIS SCD growing, the negative correlations
enhanced, and peaked when SCD is 347d, and then decreased rapidly, indicating that 347d is an optimal SCD
threshold (Figure 4). With SCD changing from 348 to 365d, the rapid decreasing of the negative correlations
could be attributed to annual cumulated errors in MODIS snow mapping algorithm and cloud removal method,
thus the high SCD (from 348 to 365d) is not suit for the threshold to estimating perennial snow cover and
SLA-EMS. Using MODIS SCD threshold as 347d, the scatter plots and linear regression parameters between
glacier annual mass balances and grid SLA-EMS (30km) for the 12 measured glaciers are shown in Figure 5 and
Table 1. They show that the 30km SLA-EMS extracted with 347d as MODIS SCD threshold has significant
linear relation with glacier annual mass balance.

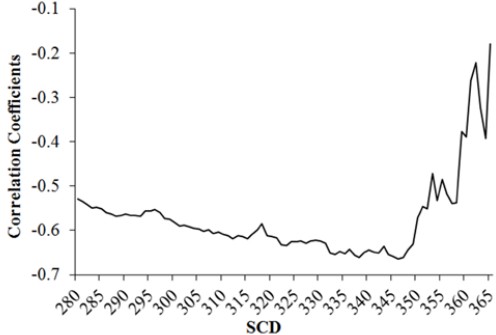


**Figure 4.**    Average correlation coefficients between annual mass balance of the 12 measured glaciers and their
corresponding grid (30km) snowline altitude at the end of melting season (SLA-EMS), changing with the MODIS
snow covered-days (SCD) threshold. The peaked negative correlations when SCD is 347d, indicating that 347d is an
optimal MODIS SCD threshold for extraction of SLA-EMS.



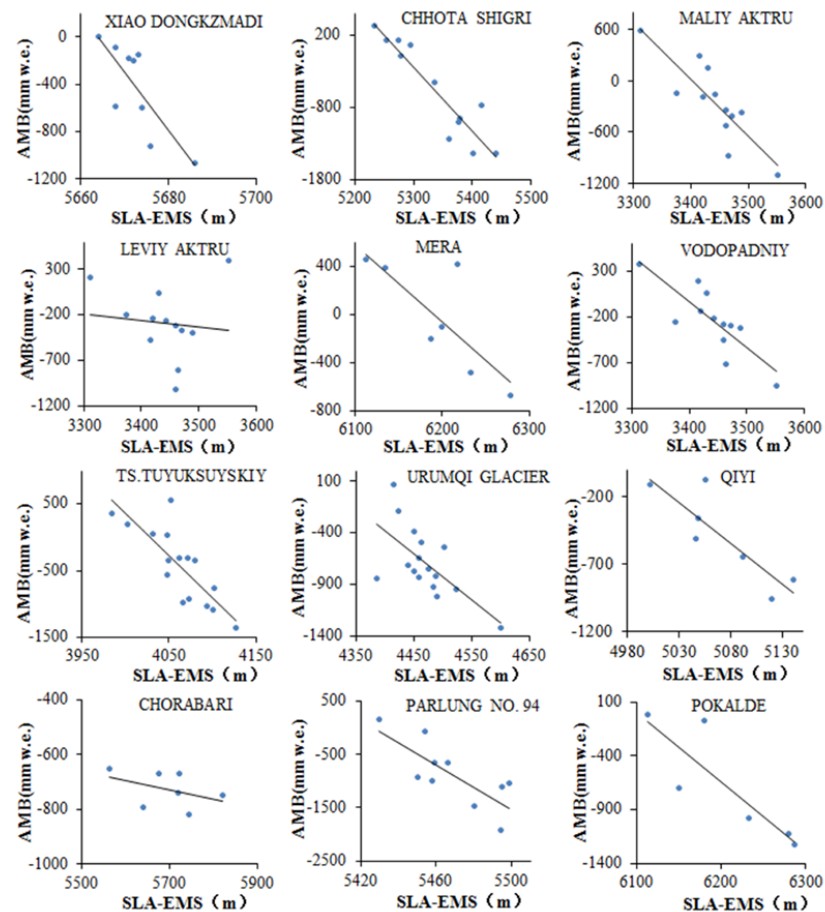

226

**Figure 5.** Scatter plots between glacier annual mass balances (AMB) and their corresponding grid (30km) snowline
altitude at the end of melting season (SLA-EMS) for the 12 measured glaciers. The MODIS SCD threshold of 347d
is used in the extraction of SLA-EMS.

**Table 1.** Linear regression parameters between annual mass balances (y) and the extracted 30km grid
SLA-EMS (x) for the 12 measured glaciers.

| Glaciers | Number of observations (yr) | Linear fitting equation | Correlation coefficients (R) |
|---|---|---|---|
| XIAO DONGKZMADI | 10 | y = -49.359x + 279566 | -0.8 |
| CHHOTA SHIGRI | 12 | y = -8.8258x + 46542 | -0.93 |
| MALIY AKTRU | 12 | y = -6.7223x + 22876 | -0.85 |
| LEVIY AKTRU | 12 | y = -0.7471x + 2283.9 | -0.11 |
| MERA | 8 | y = -6.4232x + 39766 | -0.76 |
| VODOPADNIY (NO.125) | 12 | y = -5.0136x + 17001 | -0.82 |
| TS.TUYUKSUYSKIY | 16 | y = -12.608x + 50791 | -0.76 |
| URUMQI GLACIER NO. 1 | 16 | y = -4.4547x + 19215 | -0.65 |





| | | | |
|---|---|---|---|
| QIYI | 7 | y = -6.0463x + 30172 | -0.71 |
| CHORABARI | 7 | y = -0.3482x + 1257.1 | -0.44 |
| PARLUNG NO. 94 | 10 | y = -20.797x + 112856 | -0.6 |
| POKALDE | 6 | y = -6.415x + 39131 | -0.57 |

In addition, we also use several Landsat TM/ETM+/OLI images of 5 selected Landsat scenes to determine
the MODIS SCD threshold for estimating perennial snow cover. In the process of Landsat images selection, we
focus on the principle of that the imaging time is near the end of summer and are almost unaffected by the cloud.
The specific dates and locations of Landsat ETM+/OLI images are listed in Figure 2 and Table 2. In this work,
Landsat images are firstly classified as snow or non-snow using the current "SNOWMAP" (Hall et al., 1995)
approach. And then the perennial snow cover maps (minimized snow cover) for each Landsat scenes are
produced from a composite of multi-images, in which the pixel will be identified as perennial snow cover only
if it is classified as snow in all the selected images of the scene. Finally, we compare the Landsat-derived
perennial snow cover against that of MODIS-derived in the 5 Landsat scenes, and changing with the MODIS
SCD threshold. As Figure 6 shows, although the optimal SCD thresholds (with the area ratio is equal to 1) are
varying with the regions, the average of them is also about 347d. Therefore, in this study, we ultimately chose
the MODIS SCD thresholds as 347d for estimating perennial snow cover and SLA-EMS.
**Table 2.** Information about Landsat images used in calibration of MODIS SCD threshold.

| Region | Sensor | Date | Path | Row |
|---|---|---|---|---|
| S1 | OLI | 2013/07/30、2013/08/15、2013/08/31、2013/09/16 | 148 | 31 |
| S2 | OLI | 2014/07/22、2014/08/07、2014/08/23、2014/09/08 | 151 | 33 |
| S3 | OLI | 2013/07/08、2013/07/24、2013/08/25、2013/09/10 | 149 | 35 |
| S4 | TM | 2009/06/28、2009/07/30、2009/08/15、2009/08/31 | 145 | 35 |
| S5 | ETM+ | 2002/07/13、2002/07/29、2002/08/14、2002/08/30 | 135 | 39 |

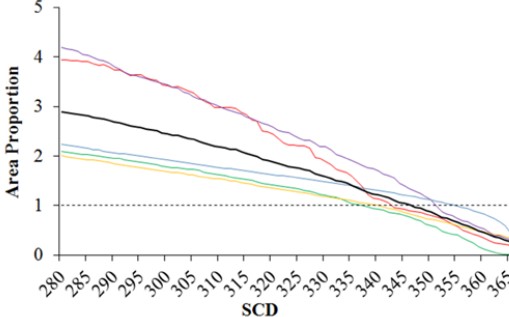


**Figure 6.** Comparisons of Landsat-derived perennial snow cover area and that of MODIS-derived in the 5 Landsat scenes
(Region S1-S5 in Figure 2 and Table 2), changing with the MODIS SCD threshold. The Y-axis represents the area ratio of
Landsat-derived perennial snow cover area divide by that of MODIS-derived, and bold black line represents the averages of
the 5 regions.
### 3.2.3. Determination Boundary Altitude Value of Perennial Snow Cover

Determination the altitude value of the SLA-EMS for every glacier grid (30km) is carried out using the area-elevation distribution curve and the perennial snow cover area. The area-elevation distribution curve for each glacier grid is generated using the resampled SRTM DEM data (500m). Take one of glacier grids as an example, the elevation measuring graph of the area-elevation distribution curve is shown in Figure 7. The altitude value of SLA-EMS for a particular year of a grid is calculated by selecting the elevation for which the area above this elevation is equal to the perennial snow cover area in the grid. Thus, the SLA-EMS in the HMA is able to be measured on a grid-by-grid basis, as long as the yearly perennial snow cover area was accurately obtained.

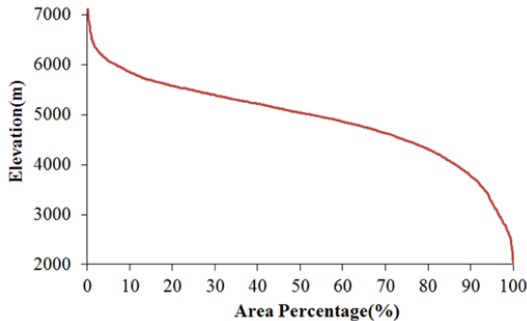

**Figure 7.** An example of area-elevation distribution curve for one of the glacier grids. The X-axis represents the percentage area above the corresponding elevations (Y-axis) in the grid.

### 3.3. Methodology of SLA-EMS Changes and the Linkages with Climate Factors

Trend analysis of a time series consists of the magnitude of trend and its statistical significance. Linear regression analysis is one of the most common methods of simulating the change trend of a time series. In this study, the ordinary least squares (OLS) regression is employed to calculate the linear trends of the SLA-EMS over the 16 years. A trend is considered to be statistically significant if its significance levels at 5%. Specifically, the slope of the least-squares line fitting of the SLA-EMS is calculated as:

$$Slope = \frac{n \times \sum_{i=1}^{n} i \times SLA_i - \sum_{i=1}^{n} i \sum_{i=1}^{n} SLA_i}{n \times \sum_{i=1}^{n} i^2 - \left(\sum_{i=1}^{n} i\right)^2} \tag{2}$$

where, Slope is the slope of the least-squares line fitting; i is the serial number from 1 to 16 for the years from 2001 to 2016; n is the cumulative number of years; and $SLA_i$ is the value of SLA-EMS in the ith year. When Slope > 0, there is an increasing tendency; when Slope = 0, there is no increasing or decreasing tendency; when Slope < 0, there is a decreasing tendency. And their significance levels (P) of F-test are presented.

Pearson correlation analysis is used to investigate the correlations between SLA-EMS, and temperature and precipitation dynamics for the 16 years (2001-2016). Pearson correlation coefficient (r) is a measure of the linear correlation between two variables x and y, which can be calculated with Equation (3)



$$r_{xy} = \frac{\sum_{i=1}^{n}(x_i - \overline{x})(y_i - \overline{y})}{\sqrt{\sum_{i=1}^{n}(x_i - \overline{x})^2}\sqrt{\sum_{i=1}^{n}(y_i - \overline{y})^2}} \qquad (3)$$

where $r_{xy}$ is the correlation coefficients between x and y; x is the SLA-EMS, y is the temperature or precipitation;
n is the number of the samples; $\overline{x}$ and $\overline{y}$ denote the average values of the x and y, respectively. Since the
correlation coefficients are determined in the period from 2001 to 2016, n equals to 16 here. The correlation is
considered to be significant if it is at the 5% significance levels.
**4. Results**
**4.1 Spatial Pattern of SLA-EMS**
For each glacier grid (30km) in the HMA, the yearly SLA-EMS has been calculated. Figure 8 presents the
spatial patterns of SLA-EMS in the HMA during 2001-2016. The spatial changes of the SLA-EMS (from
3114 to 6907m) exhibit a large spatial heterogeneity in the HMA (Figure 8). The average SLA-EMS (6091 m)
in east Himalaya is the highest among the different subregions, while the lowest (3575 m) is in Altay and
Sayan. From the whole of the HMA, the average SLA-EMS is 5256 m (Figure 9).

Generally, the SLA-EMS in the HMA decreases with increase of latitude; in southern HMA (such as the

east and central Himalayas, inner Tibet), the SLA-EMS in many grids higher than 6200m, while in northern
HMA (the Altay and Sayan), they are less than 3200m (Figure 8). Furthermore, from the high altitude region
of Himalayas and inner Tibet to surrounding low mountainous area, the SLA-EMS gradually decreases; and
there is a significant positive correlation between SLA-EMS and elevation, which indicates the spatial
patterns of the SLA-EMS are also controlled by the change of altitude (Figure 10).




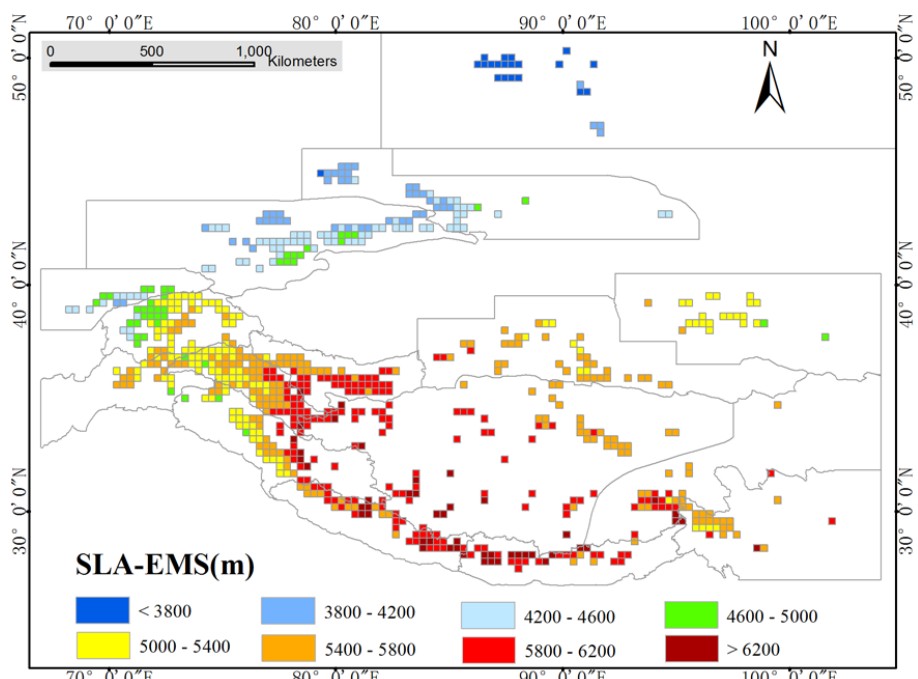

**Figure 8.** Spatial patterns of average SLA-EMS in the HMA for 2001-2016.

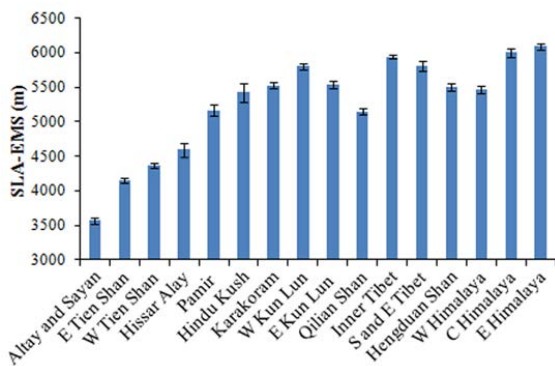

**Figure 9.** Average SLA-EMS for different subregions. Error bars show the standard deviation, indicating the interannual variations of SLA-EMS from 2001 to 2016.

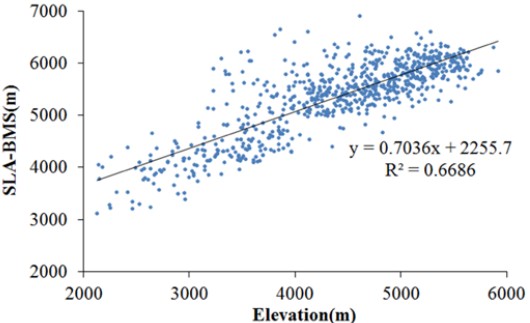





**Figure 10.** Relationships between SLA-EMS and elevations (30km). The X-axis represents the average elevation of the 30km glacier grids, Y-axis represents the extracted SLA-EMS of the glacier grids.

**4.2 Spatiotemporal Variations of SLA-EMS during 2001-2016**

Figure 11 illustrates the linear trend (*Slope*) of SLA-EMS and its significance level at the grid scale for the 16 years. On the whole, the SLA-EMS shows a rising trend in HMA, although a large number of glacier grids (40.6%) are characterized by weak trends in SLA-EMS ($-3 < Slope < 3$ m yr$^{-1}$). The linear trends for SLA-EMS in 75.3% of the grids are increased to different degrees ($0.0 < Slope < 24.5$ m yr$^{-1}$), and the increase trends in 24.2% of the grids are statistically significant ($P<0.05$). The SLA-EMS increases with the *Slope* greater than 3 m yr$^{-1}$ account for 39.4%. But only 16.1% of the grids are characterized by decrease trends in SLA-EMS ($-13.6 < Slope < 0.0$ m yr$^{-1}$), and the significant decreased ($P<0.05$) grids even less than 1%. The grids with significant increased SLA-EMS are mainly distributed in Tien Shan, east and central Himalayas, inner Tibet, and the Nyainqentanglha of south and east Tibet. While the weak trends of decrease in SLA-EMS appears in few areas of Karakoram, Pamir, Hindu Kush, and west Himalaya (Figure 11).
**Figure 11.** Change trend (*Slope*) of SLA-EMS (a); and its significance level (b) in grid scale during 2001-2016 of HMA. Significant changes indicate its statistical significance at the 5% level.

Figure 12 further present the interannual changes and linear trend (*Slope*) of SLA-EMS for different subregions. Significant increase trends of SLA-EMS during the 16 years are found in Tien Shan, inner Tibet,



south and east Tibet, east Himalaya and Hengduan Shan; especially, in southeastern of HMA (south and east
Tibet, east Himalaya and Hengduan Shan), the average SLA-EMS increment is relatively higher ($Slope \geq 7.48$
m yr$^{-1}$). While in other regions, the SLA-EMS shows nonsignificant increase or large interannual fluctuation;
the large interannual fluctuations of SLA-EMS are mainly located in western HMA (Pamir, Hindu Kush and
Hissar Alay). The SLA-EMS in the Karakoram, Pamir, Hindu Kush and west Himalaya show stability or
slightly decreasing trend.

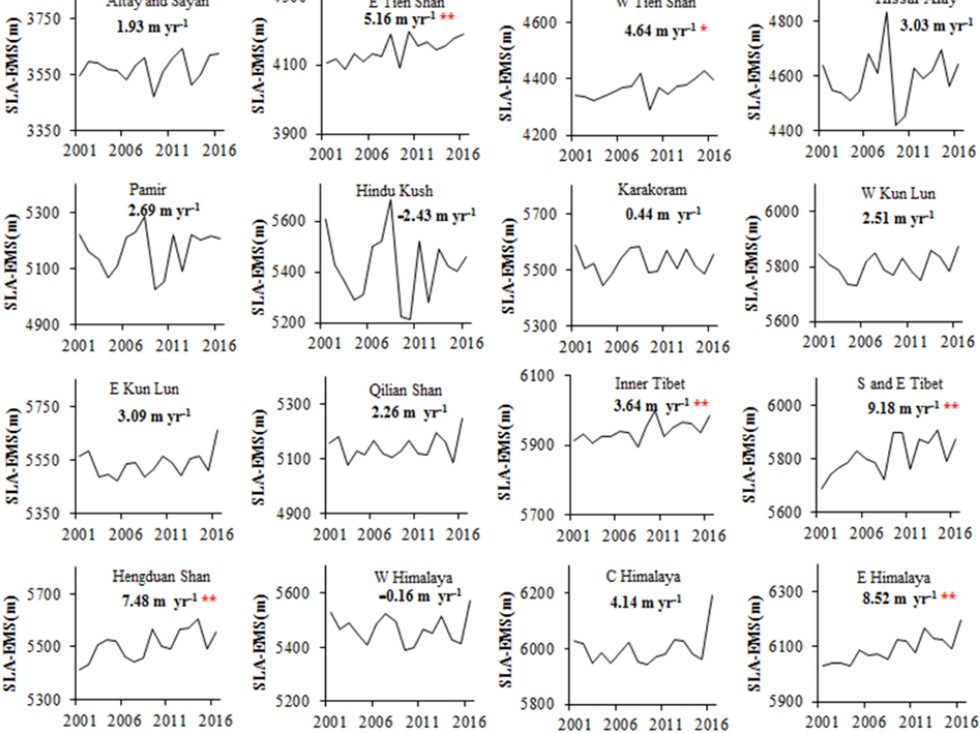

**Figure 12.** Interannual variation and linear trend (*Slope*) of SLA-EMS for different subregions of HMA from 2001 to
2016. ** and * indicate statistical significance at the 0.01 and 0.05 level, respectively.
**4.3. Correlations between SLA-EMS, Temperature and Precipitation**
To explore the possible mechanisms for SLA-EMS changes of the HMA, we examined the linkages between
the SLA-EMS and two important climate parameters (temperature and precipitation) variations by the
correlation analysis. In this study, the in situ climate data from 172 meteorological stations (which are located
in the study area) are selected to investigate the relationship between the SLA-EMS, temperature and
precipitation from the different subregions. And the temperature and precipitation are divided into two time
scales (summer and annual) in the analysis. Change trends of temperature and precipitation in these regions
are approximately represented by the stations' average value, although the meteorological stations are scarce
and unevenly distributed in some subregions. Table 3 presents the Pearson correlation coefficients between
the SLA-EMS, temperature and precipitation for the period of 2001–2016. Although the correlation
coefficients between the SLA-EMS and temperature vary in regions, indicating a large spatial heterogeneity,
the SLA-EMS in almost all of the regions shows a positive correlation with the temperature, particularly a





significant positive correlation with the summer temperature (the average R= 0.64). While, there is no obvious
correlation between SLA-EMS and precipitation. These results suggest that temperature (especially the
summer temperature) was the dominant climatic factor affecting the interannual variations of SLA-EMS in
the study area, and the variation trend and high fluctuation of SLA-EMS in the HMA (Figure 12) is mostly
due to the variability of temperature. The rising temperature in the snowmelt period can increase snow
melting and decrease snowfall, both of which can lead to a higher SLA-EMS. If the global warming trend
continues, the increase of the SLA-EMS caused by the high-positive correlation with temperature may result
in significant changes in water resources and river flows in the HMA. This will exert effects on the ecosystem,
irrigation-dependent agriculture, and domestic water in the densely populated downstream areas.
**Table 3.** Pearson correlation coefficients between the SLA-EMS, temperature, and precipitation in different subregions
for the period of 2001–2016.

| Regions | Summer temperature | Annual temperature | Summer precipitation | Annual precipitation |
|---|---|---|---|---|
| S and E Tibet | 0.69 ** | 0.72 ** | 0.42 | 0.40 |
| Hengduan Shan | 0.34 | 0.34 | 0.29 | -0.01 |
| Qilian Shan | 0.72 ** | 0.60 * | 0.19 | 0.03 |
| Inner Tibet | 0.32 | 0.13 | -0.26 | -0.23 |
| E Tien Shan | 0.60 * | 0.22 | -0.01 | 0.08 |
| W Tien Shan | 0.76 ** | 0.29 | 0.00 | -0.16 |
| E Himalaya | 0.67 ** | 0.38 | -0.14 | -0.21 |
| E Kun Lun | 0.64 ** | 0.47 | -0.09 | -0.06 |
| C Himalaya | 0.58 * | 0.53 * | 0.31 | -0.23 |
| Pamir | 0.75 ** | 0.37 | 0.02 | -0.02 |
| W Himalaya | 0.83 ** | 0.55 * | 0.01 | -0.01 |
| Altay and Sayan | 0.66 ** | 0.12 | 0.28 | 0.02 |
| Hissar Alay | 0.83 ** | 0.11 | N.A. | N.A. |
| Hindu Kush | 0.63 ** | 0.26 | N.A. | N.A. |
| W KunLun | N.A. | N.A. | N.A. | N.A. |
| Karakoram | N.A. | N.A. | N.A. | N.A. |

Note: ** and * indicate statistical significance at the 0.01 and 0.05 level, respectively. The regions without
meteorological stations (or missing data) are represented by "N.A." (meaning "not available").
**5. Discussion**
Satellite remote sensing data has been used to monitor regional snowline altitude for a long time, and visual
interpretation of Landsat MSS, TM and ETM+ (and so on) images observed in the end of summer is the
common method for investigating the SLA-EMS changes in a local areas (McFadden et al., 2011; Pandey et
al., 2013; Rabatel et al., 2012; Tawde et al., 2016; Zhang and Kang, 2017). Since the 2000s, the MODIS snow
cover products provide an excellent opportunity to study the snow cover for global or large-scale areas, and
they have been used to investigate the spatiotemporal changes of the seasonal or transient snowline altitude
(Krajčí et al., 2016; Krajčí et al., 2014; Parajka et al., 2010; Spiess et al., 2016; Tang et al., 2014; Verbyla et
al., 2017). However, there is still little suitable method to assessing large-scale SLA-EMS changes using



MODIS snow cover product. Therefore, a methodology of SLA-EMS determination from cloud-removed
MODIS snow cover products is developed in this study. The highlights of this methodology can be
summarized as follows: (1) the cloud cover in MODIS snow cover products were effectively removed; (2) in
order to accurately estimate the perennial snow cover, the MODIS SCD threshold was calibrated using both
the glaciers annual mass balance observations and Landsat images; (3) dividing glacier grids, and
determination altitude value of SLA-EMS on a grid-by-grid basis using the area-elevation distribution curve
and the perennial snow cover area. This study can also be intended as a precedent toward using MODIS snow
cover products to assess SLA-EMS at the large-scale areas for better understanding of water resources and
climatology of cold region. However, we cannot ignore the limitations of using the coarse resolution dataset
(500m); the MODIS dataset is not suitable for the extraction of SLA-EMS in small-scale area, and this is also
why the glacier grids are divided as big to 30km in the study.

Although earlier studies have shown that the snowline identified by satellite images at the end of the
glaciological year can be used as an indicator of the glacier annual mass balance (Braithwaite, 1984; Rabatel et
al., 2005, 2008, 2012; Xie et al., 1996), the glacier mass balance is presently estimated by three main methods.
Glaciological method (also known as field estimate) is one of the most commonly used method, where glacier
mass balance is estimated using in situ field studies. But, due to mountainous terrain and logistic reasons, this
method is limited to a few glaciers. The second method is the local remote sensing analysis (geodetic method
using DEM differencing). Geodetic method provided mass balance estimates only for areas smaller than a few
thousand square kilometres (Pieczonka and Bolch, 2015; Rankl and Braun, 2016; Shangguan et al., 2015) and
for varying periods, and has demonstrated that these sub-regional measurements are not representative at the
larger scale (Gardner et al., 2013). The third method is large-scale satellite measurements, such as laser
altimetry (ICESat). However, ICESat-1 laser altimeter operated only from 2003 to 2009, and had a sparse
spatial sampling leading to potential large bias (Treichler and Kääb, 2016). Therefore, it is still difficult to
achieve continuous spatial-temporal monitoring of the glacier mass balance for a large scale area. In this work,
we find that there is a good linear regression relationship between the grid (30km) SLA-EMS and annual
mass balance of glaciers in the HMA (Figure 5 and Table 1). The SLA-EMS is negatively related with the
glaciers annual mass balance (the average correlation coefficients is -0.66). These demonstrate that the
MODIS extracted grid SLA-EMS can be a good indicator of large-scale glaciers annual mass balance. The
proposed SLA-EMS determination method and SLA-EMS datasets in this research will have good potential in
reconstruction or extending the glacier annual mass balance time series at large-scale area such as the HMA.
However, the quantitative relationship between SLA-EMS and glacier annual mass balance needs to be
further studied in the future work using more abundant glacier annual mass balance data, for example using
time series of DEMs derived from ASTER optical satellite stereo-images (Brun et al., 2017).

The spatial distribution pattern of SLA-EMS in the HMA is mainly affected by two factors: latitude and
topographic altitude (Firgure 8 and 10). And the spatial pattern of SLA-EMS (Firgure 8) in this study is very
in keeping with the spatial distributions of the ELA of the monitoring glaciers in High Asia (Ye et al., 2016).
The latitudinal distribution pattern of SLA-EMS manifest as the SLA-EMS gradually decreasing from south
to north, that is Himalayas > inner Tibet > KunLun > Tien Shan > Altay and Sayan (Firgure 8). The
differences in solar radiation and temperature caused by latitude could be the main reason for latitudinal
pattern of SLA-EMS. The significant positive correlation between SLA-EMS and elevation (Firgure 10), that
is, the topographic altitude controlled spatial pattern of the SLA-EMS could be ascribed to the mass elevation



effect, which is essentially the result of the thermodynamic effect of mountain masses (Han et al., 2018; ZHANG et al., 2016), and virtually leading to higher temperature in the interior than in the outside of mountain masses at same elevation and on similar latitudes, and has been recognized as a significant contributor to the vertical distribution of mountain snowline and timberline (Han et al., 2012; Han et al., 2011).

Under the context of global warming, the cryosphere (snow cover, glaciers, glacial lakes and permafrost) of the HMA has been changing rapidly (Brun et al., 2017; Cheng and Wu, 2007; Rittger et al., 2016; Yao et al., 2012; Zhang et al., 2015). Like many glaciers worldwide, the glaciers in HMA have generally been losing mass over the past decades, but a subset of glaciers in the region have been stable or even slight growing. This peculiar behavior, first observed over the Karakoram and often named the "Karakoram anomaly" (Hewitt, 2005), has been demonstrated in the Karakoram and Pamir mountains in northwest HMA (BOLCH, et al., 2012; Gardelle et al., 2012). Recent work shows the "anomaly" of glaciers mass balances even toward the northeast of the Karakoram, in the west KunLun region (Brun et al., 2017; Lin et al., 2017). In this work, we find the SLA-EMS in HMA generally shows a rising trend in the 16 years (Figure 11 and 12), and the significant increase trends of SLA-EMS are mainly located in east Tien Shan (5.16 m yr$^{-1}$), west Tien Shan (4.64 m yr$^{-1}$), Inner Tibet (3.64 m yr$^{-1}$), south and east Tibet (9.18 m yr$^{-1}$), east Himalaya (8.52 m yr$^{-1}$), and Hengduan Shan (7.48m yr$^{-1}$). These rising trends of SLA-EMS may indicate decreases in glacial annual mass balance in the 16 years. Under the background of the generally losing glaciers mass in these areas (i.e. average annual mass balance is already negative), if the SLA-EMS continues to rise as a result of global warming, it will accelerate the negative mass balances of the glaciers. While, in the glacier mass balance anomaly areas, such as Karakoram, Pamir, Hindu Kush and west KunLun, the SLA-EMS shows no obvious trend during the examined period although a strong interannual variability is discovered.

It is true that the meteorological stations used in the correlation analysis of this study are heterogeneous and mostly located in the valley, which may not accurate represent the temperature and precipitation conditions at higher altitudes where the glaciers and perennial snow cover frequently presents. While, strong linkage between the SLA-EMS and temperature is found. That is, Temperature (especially the summer temperature) was the dominant climatic factor affecting the changes of SLA-EMS in HMA (Table 3). However, the effect of climate change on SLA-EMS is very complex. First, the other climate factors such as intensity of solar radiation, vapor pressure, wind velocity, and their synergistic effect also give rise to SLA-EMS variation, which should not be ignored. Second, climate changes and their effect on snow cover vary with geographical environment, like the regions of "Karakoram anomaly". Furthermore, the period of 16 years, is a long time considering available MODIS information, while is not sufficient for statements about climate change. Most of the trends for SLA-EMS change are not reached at the statistical significant level. A longer time series of data needs to be examined in further studies to obtain some more definitive conclusions about temporal trends of SLA-EMS and the relationship with climate change.

## 6. Conclusions

This study presents a large-scale SLA-EMS monitoring method based on the cloud-removed daily MODIS FSC data. In this method, the extent of HMA is divided into 744 glacier grids (30km); the MODIS SCD threshold for estimating perennial snow cover and SLA-EMS is deliberately calibrated using glaciers mass





balance observations and Landsat images; and the altitude value of the SLA-EMS for these glacier grids is extracted using the area-elevation distribution curve (from DEM) and the perennial snow cover area. We examine large-scale spatial patterns of the SLA-EMS across HMA and identify trends over the past 16 years, and also explore the possible linkage between SLA-EMS and temperature and precipitation. The main findings are summarized as the follows:

(1) There are good linear regression relationships (the average R = -0.66) between the grid (30km) SLA-EMS and glaciers annual mass balance over the HMA. This implies that the SLA-EMS monitoring method could help reconstruction or extending the glacier annual mass balance time series at large-scale area. However, the quantitative relationships between SLA-EMS and glaciers annual mass balance are spatially complex and will require further spatially resolved assessments and using more abundant glacier annual mass balance data.

(2) The spatial pattern of SLA-EMS across HMA is mainly affected by two factors: latitude and topographic altitude. The latitudinal pattern of SLA-EMS manifest as the gradually decreasing from south to north, that is Himalayas > inner Tibet > KunLun > Tien Shan > Altay and Sayan. Due to the mass elevation effect, the SLA-EMS decreases from the high altitude region of Himalayas and inner Tibet to surrounding low mountainous area.

(3) The SLA-EMS of HMA generally shows a rising trend in the recent years (2001-2016). In total, 75.3% (24.2% with a significant increase) and 16.1% (less than 1% with a significant decrease) of the glacier grids in HMA show increasing and decreasing trends in SLA-EMS, respectively. The SLA-EMS significant increases in Tien Shan, Inner Tibet, south and east Tibet, east Himalaya and Hengduan Shan; while there are no obvious trends in the Karakoram, Pamir, Hindu Kush, west KunLun and west Himalaya.

(4) The SLA-EMS over the HMA shows a positive correlation with the temperature, particularly a significant positive correlation with the summer temperature (R= 0.64), while no obvious correlation with precipitation. Temperature (especially the summer temperature) was the dominant climatic factor affecting the variations of SLA-EMS over the HMA. If the global warming continues, the rising of the SLA-EMS may accelerate the negative mass balances of the most glaciers over the HMA, lead to change the flow regimes and water availability, thus impacting ecosystem, agriculture and water resources in the densely populated downstream areas.

**Acknowledgments:** Many thanks to Etienne Berthier (the Editor) for insightful comments which has improved this manuscript. This study was financially supported by the National Natural Science Foundation of China (Grant No. 41871058, 41501070), Natural Science Foundation of Hunan Province, China (Grant No. 2018JJ3154), and the National Natural Science Foundation of China (Grant No. 41771075, 41701061).

**Author Contributions:** Zhiguang Tang conceived and designed the study, and wrote the paper. Zhiguang Tang and Xiaoru Wang contributed to the development of the methodology and performed the experiments. Xiaoru Wang analyzed the data. Jian Wang, Xin Wang and Junfeng Wei contributed to discussions and revisions.

**Conflicts of Interest:** The authors declare that they have no conflict of interest.



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
