# Peer review of "Investigating spatiotemporal patterns of snowline altitude at the end of melting season in High Mountain Asia, using cloud-free MODIS snow cover product, 2001-2016"

_The Cryosphere, 2019_

## Referee Comment (RC1) · Anonymous Referee #1 · 30 Jul 2019

The manuscript presents a remote sensing study, aiming to detect large-scale patterns of snowline altitude at the end of melting seasons for High Mountain Asia and the period 2001 to 2016. The proposed method is based on MODIS fractional snow cover data that are processed using a previously published routine to remove cloud cover. This method is extended by an approach to estimate perennial snow by empirically matching MODIS-derived snow-covered days to glacier mass balances and Landsat-based snow cover estimates. Strongly generalized results are then analyzed towards intrinsic trends and correlations with meteorological station data by applying basic statistic

tools, i.e. linear regression and correlation coefficients. As such, the overall topic of the manuscript is relevant and fitting TC's scope but the methods and data hardly satisfy basic standards. The actual result data is not included with the submission, so that an assessment of their quality is not possible. I find this particularly problematic since the manuscript is widely lacking critical reflections on the method, even though its weaknesses are becoming obvious in the results despite their high grade of generalization. English language of the manuscript is subject to abundant basic errors, some of which make it impossible to understand what sentences means. In summary, I find that the manuscript cannot be considered for publication owing to substantial issues in methods and over-interpretation of questionable results. I consider the work required to thoroughly tackle these issues to be way beyond the time frame for revisions, so that I unfortunately have to recommend rejection despite the interesting topic and general potential.

In the following are my comments on some of the major issues of this manuscript that might help the authors to improve the study for a future submission.

The methodological evaluation is not robust and a critical evaluation of the actual data quality is lacking: - MODIS data is presented as the 'most recent and advanced remote sensing snow product' (L76) and treated as this throughout the manuscript. The fact that there are much more recent missions providing more detailed data is ignored, the adverse effects of the extremely coarse resolution of ∼500 m is hardly considered, and studies clearly indicating that the quality of MODIS snow cover is less accurate for HMA (Rittger et al., 2013) is not mentioned. - MODIS FSC data treated with the 'cloud removal method' by Tang et al. (2016) are presented as having a 'high snow-classification accuracy' (L94). However, a considerable part of the methods focuses on finding an empirical threshold to replace the number of days per year for perennial snow cover –which should clearly be 365– with a number that fits observations. - It is generally stated that there is a 'significant linear relation' of the MODIS-derived SLA-EMS results with WGMS glacier mass balance measurements (L209f), but the

presented data clearly shows this is not the case. Only for six out of twelve glaciers 95% critical values of sample correlation are reached. Conversely, data for Leviy Aktru is basically uncorrelated and R values for Chorabari as well as Pokalde are far off the thresholds. Also, RMSE and $R^2$ values are not considered at all. - Using Landsat data to evaluate the MODIS snow cover results is a good idea. However, instead of simply checking whether the overall areas are equal I would consider it mandatory to investigate in how far the classifications match.

SLAs are substantially influenced by local topography, particularly slope aspect. The 500 km input data is hardly capable of detecting these, the 30 km grid resolution pixels completely ignores such effects. Therefore, the relevance of the findings regarding (individual) glaciers is highly questionable.

It does not become clear how data from the meteorological stations is treated, i.e. how averages are calculated over space and time, and how these are related to the results. As visible in Fig. 1, meteorological stations are extremely scarce in glaciated regions. There are no data at all for the Karakoram and the W Kunlun and most of the other stations are far away from glaciers, typically in completely different climate regimes down in the valleys – this is obviously known to to the author's according to L410ff. So why not use more appropriate data, such as the freely available ERA5?

Honestly, I don't understand why the manuscript focuses on the end-melting season. By this approach the greatest advantage of MODIS data, its high temporal resolution, is lost whereas the challenges regarding the quality of specific measurements are fully affecting the results.

Large parts of the discussion do not present content supposed to be there. While interpretations of the results and evaluations of their implications against the literature are short and remain superficial, many paragraphs basically repeat statements regarding the relevance of such remote sensing studies as well as the general, subjective praise of the method.

Recommended literature: Rittger, K., Painter, T. H. and Dozier, J.: Assessment of methods for mapping snow cover from MODIS, Advances in Water Resources, 51, 367–380, doi:10.1016/j.advwatres.2012.03.002, 2013.

---

## Referee Comment (RC2) · Mauri Pelto (Referee) · 1 Aug 2019

The use of MODIS imagery to identify end of melt season snowline line altitude (SLA) as a proxy for ELA is not a new process. The MODIS derived SLA can be a proxy for mass balance, but is difficult to actually expect the approach here at the grid cell level using 500 m pixels to yield accurate mass balance values. As a consistently observable and reportable metric of glacier-climate across the HMA there is value in that this is a repeatable measurement, how accurately it predicts mass balance is not the value here. There are several significant issues the authors need to address to

make this a useful contribution. 1) There are three key references just published in The Cryosphere in the last seven years that are essential to review. All use approaches that overlap with part of the process used here. 2) The footprint of MODIS in steep terrain leads to significant errors in establishing a SLA. This has to be explored, particularly given a grid scale measurement is used, and is not validated against more specific glacier by glacier observation for a few sample grids. 3) The interpretation of the SLA variation being dependent on latitude and regional mean elevation does not capture key drivers of this including differences in moisture sources, and seasonal distribution of precipitation.

10: Why use a new acronym instead of the excepted terminology of transient snow line (TSL) for observations of the snow line not at the end of the melt season or snow line altitude (SLA) if it is the end of the melt season and is equivalent to the ELA, than just use ELA.

45: Check, Flint (1971) not a good interpretation of snowline.

56: There are useful references here including from the HMA that illustrate more frequent SLA observation (Das and Chakraborty, 2015)

79: This is not true, note work of Barundun et al (2018): "The integration of TSL observations into conventional modelling is shown to be highly beneficial for filling the gaps in long-term SMB series for periods for which direct glaciological measurements were discontinued or are missing completely."

84: Shea et al (2013) used MODIS for regional snowline altitude assessment just as you propose. There basic approach is "We describe a method to calculate regional snow line elevations and annual equilibrium line altitudes (ELAs) from daily MODIS imagery (MOD02QKM) on large glaciers and icefields in western North America. An automated cluster analysis of the cloud-masked visible and near-infrared bands at 250m resolution is used to delineate glacier facies (snow and ice) for ten glacierized regions between 2000–2011. For each region and season, the maximum observed value of

the 20th percentile of snowcovered pixels is used to define a regional ELA proxy."

87: Pelto et al (2011) compared MODIS and Landsat for snow line identification: "The MODIS imagery is from band 1 which has a resolution of 250 m. With the average surface slope of 1.6âŮę this yields an error of less than ±10 m in elevation for TSL. A comparison of a Landsat image and MODIS image from 29 July 2009 is provided (Fig. 2). It is evident that though some detail is lost the TSL position identified overall is not significantly different."

129: Given the 500 m pixel size and the average slopes how much accuracy is there for SLA? This is a key issue given you are reporting a grid cell average. This should be validated for a few particular grid cells with Landsat, SPOT or Sentinel observation of SLA in that same grid cell on glaciers. This is done for SCA using Landsat that is a different measure.

142: How many report ELA observations?

189: "However, the snow area with MODIS SCD≥365d fails to really indentify the perennial snow area, due to the affect of the annual cumulated errors in MODIS snow mapping algorithm and cloud removal method."

201: Why is the 25km2 glacier area of the 30km2 grid chosen?

214: The correlation from 332d to 347d is relatively consistent indicating this is a good window, and 347d alone does not have to be relied upon if imagery is poor. If the this time of 347 days shifts that is a measure too.

218: Given that many of these glaciers report ELA to WGMS, which provides a more direct measure of the adequacy of your method, it would be appropriate to provide this measure. The mass balance provides a correlation that is similar, however, the standard deviation between the methods is meaningless with the different units. This could be done collectively versus glacier by glacier.

241: In figure 6 it is provide legend for the various regions. The continued declining

ratio of Landsat/MODIS beyond 347d suggests that the melt season is continuing and the SLA would still be rising.

258: This method appears to be quite useful for providing a comparable SLA elevation across the region annually, even if it is not overly accurate to a glacier in particular grid cells or for mass balance assessment.

287: How does fit with the results of Barundun et al (2018) from the Tien Shan and Pamir-Altay?

288: How much of this is latitude versus level of maritime climate influence, or degree to which the glacier is a summer accumulation type? Many references have examined this issue.

291: Both altitude and elevation are used in this sentence, are you referring to the average elevation of the grid cell?

306-307: Your percentages refer to discrete linear responses and should be grouped in a way that the 100% is reached in a clear way. Lump together the significance groupings. Lump together the slope groupings.

311-313: Quantify what % of decreasing trends are from the indicated regions.

347: The comment about the rising SLA and water resources is a generic statement that needs support or removal. Thayyen and Gergan (2010) have described how the runoff from summer accumulation type glaciers is less of a resource than for other areas. If the melt season expands into the fall months as has been noted, this is a lower flow period and water resources could be increased with more glacier melt.

369: I do not see how this study is a precedent for using MODIS for snow cover mapping regardless of region or end product.

390: That SLA is a good indicator of mass balance is well established. In this case an indicator is not a substitute for any of the other methods that provide an actual quantity

that can be validated with one of the other methods.

397: The stipulation that elevation and latitude are the key variables is not supported by much of the literature that indicates how impacted by the summer monsoon and the winter westerlies is a key variable depending on location. That they are two well correlated parameters is accurate.

420: They do indicate declining mass balance.

467: The rising snowlines have already led to a decline in mass balance and mass flux down glacier. This is a continuation of regional mass loss that has driven thinning and a slowdown in glacier movement in 9 of 11 regions in HMA from 2000-2017 (Dehecq et al 2019).

Barandun, M., Huss, M.; Usubaliev, R.; Azisov, E.; Berthier, E.; Kääb, A.; Bolch, T. and Hoelzle, M. Multi-decadal mass balance series of three Kyrgyz glaciers inferred from modelling constrained with repeated snow line observations, The Cryosphere, 2018, 12, 1899-1919, https://doi.org/10.5194/tc-12-1899-2018.

Das, S, and Chakraborty, M: Delineation of glacial zones of Gangotri and other glaciers of Central Himalaya using RISAT-1 C-band dual-pol SAR. International Journal of Remote Sensing 36(6):1529-1550, DOI: 10.1080/01431161.2015.1014972, 2015

Dehecq, A., N. Gorumelon, A. Gardner, F. Brun, D. Goldberg, P. Nienow, E. Berthier, C. Vincent, P. Wagnon, and E. Trouve, 2019: Twenty-first century glacier slowdown driven by mass loss in High Mountain Asia. Nature Geoscience 12, 22–27.

Pelto, M.: Utility of late summer transient snowline migration rate on Taku Glacier, Alaska, The Cryosphere, 5, 1127-1133, https://doi.org/10.5194/tc-5-1127-2011, 2011.

Shea, J. M., Menounos, B.; Moore, R. D. and Tennant, C. An approach to derive regional snow lines and glacier mass change from MODIS imagery, western North America. The Cryosphere, 2013, 7, 667-680, https://doi.org/10.5194/tc-7-667-2013.

Thayyen, R. J. and Gergan, J. T.: Role of glaciers in watershed hydrology: a preliminary study of a "Himalayan catchment", The Cryosphere, 4, 115-128, https://doi.org/10.5194/tc-4-115-2010, 2010.

---

## Author Comment (AC1) · 5 Sep 2019

The manuscript presents a remote sensing study, aiming to detect large-scale patterns of snowline altitude at the end of melting seasons for High Mountain Asia and the period 2001 to 2016. The proposed method is based on MODIS fractional snow cover data that are processed using a previously published routine to remove cloud cover. This method is extended by an approach to estimate perennial snow by empirically matching MODIS-derived snow-covered days to glacier mass balances and Landsat-based snow cover estimates. Strongly generalized results are then analyzed towards intrinsic trends and correlations with meteorological station data by applying basic statistic tools, i.e. linear regression and correlation coefficients. As such, the overall topic of the manuscript is relevant and fitting TC's scope but the methods and data hardly satisfy basic standards. The actual result data is not included with the submission, so that an assessment of their quality is not possible. I find this particularly problematic since the manuscript is widely lacking critical reflections on the method, even though its weaknesses are becoming obvious in the results despite their high grade of generalization. English language of the manuscript is subject to abundant basic errors, some of which make it impossible to understand what sentences means. In summary, I find that the manuscript cannot be considered for publication owing to substantial issues in methods and over-interpretation of questionable results. I consider the work required to thoroughly tackle these issues to be way beyond the time frame for revisions, so that I unfortunately have to recommend rejection despite the interesting topic and general potential.

**Response:** We have made two major improvements in the revised manuscript. (1) Added the Evaluation of MODIS-derived grid (30km) SLA-EMS. (2) We have changed the temperature and precipitation data of meteorological stations into ERA5 reanalysis data, the correlation coefficients between the SLA-EMS, temperature, and precipitation during 2001 to 2016 are calculated on a grid-by-grid basis.

In addition, the English language will be further improved in the revised manuscript. The result

data (MODIS-derived SLA-EMS in the 16 years) will be available online as the supplementary data related to this article, after it is accepted.

In the following are my comments on some of the major issues of this manuscript that might help the authors to improve the study for a future submission.

The methodological evaluation is not robust and a critical evaluation of the actual data quality is lacking: - MODIS data is presented as the 'most recent and advanced remote sensing snow product' (L76) and treated as this throughout the manuscript. The fact that there are much more recent missions providing more detailed data is ignored, the adverse effects of the extremely coarse resolution of 500 m is hardly considered, and studies clearly indicating that the quality of MODIS snow cover is less accurate for HMA (Rittger et al., 2013) is not mentioned.

Rittger, K., Painter, T. H. and Dozier, J.: Assessment of methods for mapping snow cover from MODIS, Advances in Water Resources, 51, 367–380, 2013.

**Response:** In this research, we mainly focus on using MODIS snow cover products to monitor the spatial and temporal patterns of the SLA-EMS in the whole of the HMA. This research is a practice without precedent, despite the coarse resolution of the SLA-EMS dataset (30km) and the data source (500m).

The spatial resolution of the remote sensing data source should vary with the spatial scale of the research object, not always the higher the better. For example, the applications of the Landsat data and other high resolution remote sensing images in the snowline monitoring are mostly limited to individual glacier or small areas (L76), due to the low temporal resolution and large cloud cover. But the spatiotemporal pattern of SLA-EMS from the whole of the HMA under climate change is poorly understood. Thus we experimented with the cloud-removed MODIS fractional snow cover product to investigate the spatiotemporal pattern of SLA-EMS in the HMA from the grid scale of 30km. In this study, we pay attention to the spatiotemporal change of SLA-EMS on the large scale, rather than the details on the local area (such as local slope and aspect). The coarse resolution of the 500m in MODIS can be tolerated in this study.

We have checked the study of Rittger et al. (2013). It is not true for Reviewer's comment of "the quality of MODIS snow cover is less accurate in HMA". Instead, the Rittger et al. (2013) showed that the MOD10A1 fractional snow cover product in HMA were more accurate than the other areas (the Table 3). The MOD10A1 fractional snow cover product is exactly what we used in this manuscript. We appreciate the Reviewer for the recommendation of this important reference, and we have added it in the revised manuscript.

**6.1. Regional sensitivity**

As presented in Section 5, the Himalaya region has the greatest errors in snow-covered area, while the three North American regions show similar results. In North America at the 15% threshold, all binary statistics exceed 0.9, whereas in the Himalaya *Recall* drops below 0.8 and the *F* statistic therefore also drops below the North American values (Table 2).

For the fractional methods (Table 3), the RMSE for MOD10A1 is actually better in the Himalaya than in North America (0.155 vs 0.229+). MODSCAG's RMSE in the Himalaya is slightly worse than

(Rittger et al., 2013)

**Table 3**
Summary of fractional statistics.

| Region | Statistic | MOD10A1 fractional |
|---|---|---|
| Sierra Nevada | RMSE[a] | 0.276 |
| | Mean difference[b] | 0.180 |
| | Median difference | 0.190 |
| Upper Rio Grande | RMSE | 0.229 |
| | Mean difference | 0.178 |
| | Median difference | 0.193 |
| Cold Land Processes Experiment | RMSE | 0.247 |
| | Mean difference | 0.186 |
| | Median difference | 0.190 |
| Himalaya | RMSE | 0.155 |
| | Mean difference | −0.087 |
| | Median difference | −0.085 |
| All | RMSE | 0.227 |
| | Mean difference | 0.114 |
| | Median difference | 0.122 |

[a] RMSE is defined in Eq. (11).
[b] Difference is defined as of $f_{SCA}^{MODIS} - f_{SCA}^{Landsat}$.

(Rittger et al., 2013)

In section 2.2.1. MODIS Fractional Snow Cover (FSC) Data

"...*Evaluation studies have proved a high accuracy (with a mean absolute error less than 0.1) of the MODIS FSC data (Hall and Riggs, 2007; Rittger et al., 2013; Salomonson and Appel, 2004; Tang et al., 2013).*"

We agree with the Reviewer that it is necessary to evaluate the quality of the MODIS-derived SLA-EMS. We have selected 5 grids to evaluate the MODIS-derived SLA-EMS, using Landsat images (TM, ETM+ and OLI) selected from melt seasons. And the discussion for the possible uncertainty and error sources of the method are added in the Section 5. (Discussion).

[Figure]

**Figure 2.** *…The 5 blue grids (G1-G5) indicate the site for SLA-EMS evaluation.*

**Table 2.** *Information about Landsat TM/ETM+/OLI images used in validation of MODIS-derived SLA-EMS.*

| Year | G1 and G2 Path 147, Row 31 | G3 Path 151, Row 33 | G4 Path 150, Row 34 | G5 Path 145, Row 35 |
|---|---|---|---|---|
| 2001 | Aug 13 | Jul 26 | Jul 3, Aug 20, Sep 5 | Aug 25, Sep 2 |
| 2002 | Jul 1, Jul 17, Aug 18 | Aug 30 | Aug 7, Aug 23 | Jul 3 |
| 2003 | Jul 20 | Jul 16, Sep 2 | | Jul 22 |
| 2004 | Jul 22, Aug 7, Aug 23 | Jul 2, Aug 3, Sep 4 | Jul 11, Jul 27, Aug 12 | Jul 8, Aug 25, Sep 10 |
| 2005 | Jul 25, Aug 10, Aug 26 | Jul 21, Aug 6, Sep 7 | Jul 14, Aug 15, Aug 31 | Jul 11, Aug 12, Sep 13 |
| 2006 | Jul 28, Aug 13, Aug 29 | Jul 24, Aug 25, Sep 10 | Jul 17, Aug 2, Sep 3 | Jul 30, Aug 15, Sep 31 |
| 2007 | Aug 8, Aug 24, Sep 1 | Jul 27, Aug 12, Aug 28 | Jul 4, Jul 20, Aug 5 | Aug 10, Aug 18, Sep 3 |
| 2008 | Aug 2, Aug 10, Sep 3 | Jul 29, Aug 14, Aug 30 | Jul 6, Aug 7, Aug 23 | Jul 3, Aug 4, Aug 5 |
| 2009 | Jul 20, Aug 10, Aug 21 | Jul 16, Aug 1, Aug 17 | Jul 9, Jul 25, Aug 2 | Jul 6, Jul 15, Aug 31 |
| 2010 | Jul 15, Jul 23, Aug 16 | Jul 19, Aug 20, Sep 5 | Jul 12, Aug 21, Aug 29 | Jul 9, Jul 17, Aug 26 |
| 2011 | Jul 26, Aug 3, Aug 19 | Jul 22, Aug 23, Sep 8 | Jul 7, Aug 8, Aug 24 | Jul 28, Aug 5, Aug 21 |
| 2012 | Jul 28, Aug 29 | Jul 9, Aug 25, Sep 10 | Jul 17, Aug 2, Aug 18 | Aug 31 |
| 2013 | Jul 23, Jul 31, Sep 1 | Jul 19, Jul 27, Aug 28 | Jul 4, Jul 28, Sep 6 | Jul 17, Aug 2, Aug 10 |
| 2014 | Jul 10, Jul 26, Aug 11 | Jul 14, Aug 7, Aug 31 | Jul 15, Jul 23, Aug 1 | Jul 28, Aug 13, Aug 21 |
| 2015 | Aug 6, Aug 14, Aug 30 | Jul 17, Aug 18, Sep 3 | Aug 19, Aug 27, Sep 4 | Jul 23, Aug 8, Sep 9 |
| 2016 | Aug 8, Aug 24, Sep 1 | Jul 27, Aug 4, Aug 28 | Jul 20, Jul 28, Aug 13 | Jul 2, Jul 17, Jul 25 |
| Total | 43 | 43 | 44 | 41 |

In the secsion 4. Results

"*4.1 Evaluation of MODIS-derived grid (30km) SLA-EMS*

 *To evaluate our method, we compared SLA-EMS of 5 grids manually digitized from*

*high-resolution Landsat images (Table 2) with the automatic measured results from MODIS, during 2001-2016. To be consistent with the DEM data sources of the MODIS-derived SLA-EMS, the 90m SRTM DEM is also used to calculate the SLA-EMS derived from the manual delineation. For each grid and year, the highest snowline is manually digitized as the "truth-value" of SLA-EMS by combining several Landsat images of melt season. The mean absolute error, root mean square error and correlation coefficient are employed to evaluate the reliability of the MODIS-derived SLA-EMS (Table 3). In the 5 validation grids, the mean absolute error of MODIS-derived SLA-EMS compared with the manually-derived (Landsat) values is between 44.9 and 124.7 m, and the RMSE is between 52.3 and 133.4m. Despite these differences between the MODIS-derived SLA-EMS and that from manually-derived (Landsat), the correlation coefficients between them are high (between 0.63 and 0.87), and they are all significant at the 0.01 level. The significant correlations indicate that the proposed method can be used to accurately monitor the interannual variations of SLA-EMS. We believe that the MODIS-derived SLA-EMS with such accuracy in the 30km grids can be applied to investigating the spatiotemporal patterns of SLA-EMS in the HMA."*

**Table 3.** *Comparison of SLA-EMS derived from manual delineation of snowline (Landsat images) and automatic calculation from MODIS, during 2001 to 2016.*

| Site | Mean absolute error (m) | Root mean square error (m) | Correlation coefficient |
|------|------------------------|----------------------------|-------------------------|
| G1 | 76.4 | 84.4 | 0.70** |
| G2 | 124.7 | 133.4 | 0.65** |
| G3 | 67.1 | 78.0 | 0.87** |
| G4 | 50.6 | 62.2 | 0.76** |
| G5 | 44.9 | 52.3 | 0.64** |

*** indicate statistical significance at the 0.01 level.*

In the section 5. Discussion

…

"*Due to the coarse resolution (500m) of the MODIS data, the proposed method for SLA-EMS monitoring is limited to large scale areas, and it is also why the glacier grids are divided as big as 30km in this study. The uncertainty of the MODIS-derived SLA-EMS may come from different sources of errors: (1) errors occurred due to the pixel size of the remote sensing images, slope and aspect of the terrain, the accuracy of the georeferencing and the quality of the DEM (Rabatel et al., 2002, 2005, 2012); (2) the errors in MODIS snow mapping algorithm (Hall and Riggs, 2007; Rittger et al., 2013) and cloud removal method (Tang et al., 2013), although the MODIS SCD threshold is calibrated in this method.*"

MODIS FSC data treated with the 'cloud removal method' by Tang et al. (2016) are presented as having a 'high snow classification accuracy' (L94). However, a considerable part of the methods focuses on finding an empirical threshold to replace the number of days per year for perennial snow cover –which should clearly be 365– with a number that fits observations.

**Response:** Theoretically, the SLA-EMS can be determined by the MODIS derived snow covered days (SCD), that is, the boundary altitude of perennial snow cover (where the SCD≥365d).

However, the SCD≥365d is too strict and idealistic, due to the annual cumulated errors of MODIS snow mapping algorithm and cloud removal method. That is to say, for the cloud-removed MODIS snow cover images will fails to really identify perennial snow cover, as long as there is more than one error in the 365 days. Therefore, in order to accurately estimate the perennial snow cover (minimize the annual cumulated errors in MODIS snow mapping algorithm and cloud removal method), the MODIS SCD threshold was calibrated using both the glaciers annual mass balance observations and Landsat images.

It is generally stated that there is a 'significant linear relation' of the MODIS-derived SLA-EMS results with WGMS glacier mass balance measurements (L209f), but the presented data clearly shows this is not the case. Only for six out of twelve glaciers 95% critical values of sample correlation are reached. Conversely, data for Leviy Aktru is basically uncorrelated and R values for Chorabari as well as Pokalde are far off the thresholds. Also, RMSE and $R_2$ values are not considered at all.

**Response:** Many previous studies (Braithwaite, 1984; Barandun et al., 2018; Rabatel et al., 2005, 2008, 2012; Shea et al., 2013; Xie et al., 1996), which focused on the individual glacier or local areas have shown that glacier annual mass balance is highly correlated with the SLA-EMS. This is the reason why we used the glaciers annual mass balance observations to calibrate the MODIS SCD threshold, the highest negative correlations between annual mass balance and SLA-EMS indicate an optimal SCD threshold (Figure 4).

   It is normal that the correlations between glaciers annual mass balances and the MODIS-derived 30km grid SLA-EMS are not all significant (95%), due to the individual glacier annual mass balance observation is difficult to represent the true value for the 30km grid regions. How accurately the MODIS-derived SLA-EMS predicts annual mass balance is not the objective at this stage of our research. The quantitative relationship between SLA-EMS and glacier annual mass balance at the grid scale (30km) needs to be further studied in the future work using adequate spatially resolved glacier annual mass balance data, for example using time series of DEMs derived from ASTER optical satellite stereo-images (Brun et al., 2017).

   We modified the relevant text to make it more rigorous. For example, the "significant" has changed as "good", in L219.

- Using Landsat data to evaluate the MODIS snow cover results is a good idea. However, instead of simply checking whether the overall areas are equal I would consider it mandatory to investigate in how far the classifications match.

**Response:** We believe the "checking whether the overall areas are equal" is exactly a good indicator for "investigate in how far the classifications match".   In the method, the altitude value of the SLA-EMS for each glacier grid (30km) is also measured by the perennial snow cover areas and the area-elevation distribution curve (Figure 7). Therefore, we focus on the comparisons of Landsat-derived perennial snow cover area and that of MODIS-derived (checking whether the overall areas are equal) in the calibration of MODIS SCD threshold using Landsat data (Figure 6).

SLAs are substantially influenced by local topography, particularly slope aspect. The 500 km

input data is hardly capable of detecting these, the 30 km grid resolution pixels completely ignores such effects. Therefore, the relevance of the findings regarding (individual) glaciers is highly questionable.

**Response:** In this study, we pay attention to the spatiotemporal change of SLA-EMS on the large scale area, rather than the details on the local topography (such as local slope and aspect). And there are no relevant findings in this research to regarding individual glaciers. In Figure 4 and 5, we just used the glaciers annual mass balance observations to calibrate the MODIS SCD threshold, based on the findings of the previous studies (Braithwaite, 1984; Barandun et al., 2018; Rabatel et al., 2005, 2008, 2012; Shea et al., 2013; Xie et al., 1996) (they showed the glacier annual mass balance is highly correlated with the SLA-EMS).

It does not become clear how data from the meteorological stations is treated, i.e. how averages are calculated over space and time, and how these are related to the results. As visible in Fig. 1, meteorological stations are extremely scarce in glaciated regions. There are no data at all for the Karakoram and the W Kunlun and most of the other stations are far away from glaciers, typically in completely different climate regimes down in the valleys – this is obviously known to to the author's according to L410ff. So why not use more appropriate data, such as the freely available ERA5?

**Response:** We agree with the Reviewer. We have changed the temperature and precipitation data of meteorological stations into ERA5 reanalysis data. The product of "ERA5 monthly averaged data on single levels from 1979 to present" was downloaded from https://cds.climate.copernicus.eu/cdsapp#!/search?type=dataset&text=ERA5. Horizontal resolution: 0.25°×0.25°; temporal resolution: monthly.

For each grid, the correlation coefficients between the SLA-EMS, temperature and precipitation during 2001 to 2016 are calculated (Fig. 13). In Fig. 13, both of the climate variables are calculated from summer and hydrological year (from September of the previous year to August of the current year).

The relevant result analysis in this manuscript will be modified accordingly in the revised manuscript.

[Figure]

*Figure 13. The correlation coefficients between the SLA-EMS, temperature, and precipitation during 2001 to 2016.*

*Table. The averages of Pearson correlation coefficients between the SLA-EMS, temperature, and precipitation for different subregions in the period of 2001-2016.*

| Regions | Summer temperature | Annual temperature | Summer precipitation | Annual precipitation |
|---|---|---|---|---|
| S and E Tibet | 0.55 * | 0.38 | -0.05 | -0.09 |
| Hengduan Shan | 0.13 | 0.01 | 0.06 | 0.01 |
| Qilian Shan | 0.75 ** | 0.61 * | 0.26 | 0.17 |
| Inner Tibet | 0.52 * | 0.36 | -0.08 | -0.19 |
| E Tien Shan | 0.58 * | 0.17 | -0.18 | -0.16 |
| W Tien Shan | 0.55 * | 0.11 | 0.05 | -0.13 |
| E Himalaya | 0.46 | 0.23 | 0.03 | 0.01 |
| E Kun Lun | 0.64 ** | 0.48 | 0.15 | 0.14 |
| C Himalaya | 0.37 | 0.36 | 0.13 | -0.16 |
| Pamir | 0.77 ** | 0.41 | 0.04 | -0.53 * |
| W Himalaya | 0.64 ** | 0.52 * | -0.02 | -0.40 |
| Altay and Sayan | 0.67 ** | 0.30 | -0.07 | -0.32 |
| Hissar Alay | 0.67 ** | 0.13 | 0.15 | -0.61 * |
| Hindu Kush | 0.86 ** | 0.46 | -0.36 | -0.68 ** |
| W KunLun | 0.63 ** | 0.22 | -0.05 | -0.08 |
| Karakoram | 0.57 * | 0.32 | 0.19 | -0.28 |

Honestly, I don't understand why the manuscript focuses on the end-melting season. By this approach the greatest advantage of MODIS data, its high temporal resolution, is lost whereas the challenges regarding the quality of specific measurements are fully affecting the results.

**Response:** The reasons for why the manuscript focuses on the snowline altitude at the end-melting season (SLA-EMS) were clearly introduced in the introduction.

*"The snowline altitude at the end of melting season (SLA-EMS) approximates the equilibrium line altitude (ELA), it can serves as a good proxy for ELA and therefore for the mass balance of glaciers (McFadden et al., 2011; Pandey et al., 2013; Rabatel et al., 2005, 2012; Tawde et al., 2016). Numerous studies (Braithwaite, 1984; Barandun et al., 2018; Rabatel et al., 2005, 2008, 2012; Shea et al., 2013; WGMS, 1991-2013; Xie et al., 1996) have shown that glacier annual mass balance is highly correlated with the ELA or SLA-EMS, and the SLA-EMS enables reconstruction of annual mass balance time series. The climate sensitivity of SLA-EMS has been generally emphasized as a supplement to current climate change indicator systems. A study of the spatial-temporal variations of the SLA-EMS can help in assessing the hydrologic cycle balance as well as to understand the regional and global cryosphere and climate changes."*

Large parts of the discussion do not present content supposed to be there. While interpretations of the results and evaluations of their implications against the literature are short and remain superficial, many paragraphs basically repeat statements regarding the relevance of such remote sensing studies as well as the general, subjective praise of the method.

**Response:** We will try to improve it in the revised manuscript.

---

## Author Comment (AC2) · 5 Sep 2019

Mauri Pelto (Referee)

mauri.pelto@nichols.edu

The use of MODIS imagery to identify end of melt season snowline line altitude (SLA) as a proxy for ELA is not a new process. The MODIS derived SLA can be a proxy for mass balance, but is difficult to actually expect the approach here at the grid cell level using 500 m pixels to yield accurate mass balance values. As a consistently observable and reportable metric of glacier-climate across the HMA there is value in that this is a repeatable measurement, how accurately it predicts mass balance is not the value here. There are several significant issues the authors need to address to make this a useful contribution. 1) There are three key references just published in The Cryosphere in the last seven years that are essential to review. All use approaches that overlap with part of the process used here. 2) The footprint of MODIS in steep terrain leads to significant errors in establishing a SLA. This has to be explored, particularly given a grid scale measurement is used, and is not validated against more specific glacier by glacier observation for a few sample grids. 3) The interpretation of the SLA variation being dependent on latitude and regional mean elevation does not capture key drivers of this including differences in moisture sources, and seasonal distribution of precipitation.

**Response:** It is true that there have been studies using MODIS imagery to identify summer transient snowline (Pelto, 2011) or maximum value of snowline line altitude at the ablation season (Shea, 2013). However, we think, there are obvious differences in the method of this manuscript; because it is more efficient in monitoring of spatial and temporal continuous SLA-EMS over a large-scale area (such as the whole of the HMA), and is a repeatable measurement with more automatic (no need to manually select the less-cloud images). The detailed explanation sees the the **response to the issues 1**). The highlights of the methodology in this can be summarized as follows: (1) the cloud cover in daily MODIS snow cover products were effectively removed; (2) using MODIS extracted snow covered days (SCD) to estimate the perennial snow cover, and the MODIS SCD threshold was calibrated using both the glaciers annual mass balance observations

and Landsat images; (3) dividing glacier grids, altitude value of SLA-EMS was calculated on a grid-by-grid basis using the area-elevation distribution curve and the perennial snow cover area.

We agree with the Reviewer that "as a consistently observable and reportable metric of glacier-climate across the HMA there is value in that this is a repeatable measurement, how accurately it predicts mass balance is not the value here." The main objectives of this research are to: propose a method for spatially resolved estimation of SLA-EMS over a large-scale area, based on the cloud-removed daily MODIS FSC data; and give detailed estimates of the changes of SLA-EMS in HMA during 2001-2016 on a grid-by-grid (30km x30km) basis, and the possible cause for the SLA-EMS spatiotemporal changes.

Many other previous studies (Braithwaite, 1984; Barandun et al., 2018; Rabatel et al., 2005, 2008, 2012; Shea et al., 2013; WGMS, 1991-2013; Xie et al., 1996) have shown that glacier annual mass balance is highly correlated with the ELA or SLA-EMS, and the SLA-EMS is enables reconstruction of annual mass balance time series. Therefore, we consider the MODIS-extracted SLA-EMS grid datasets (30km x30km) in this research will have potential in reconstruction or extending the glacier annual mass balance time series for large-scale area of the HMA. However, the quantitative relationships between SLA-EMS and glacier annual mass balance at the scale of 30km grids needs to be further studied in the future work using adequate spatially resolved glacier annual mass balance data, for example using time series of DEMs derived from ASTER optical satellite stereo-images.

In the following sections, we try to address the issues point-by-point with changes in the manuscript.

**1)** There are three key references just published in The Cryosphere in the last seven years that are essential to review. All use approaches that overlap with part of the process used here.

Barandun, M., Huss, M.; Usubaliev, R.; Azisov, E.; Berthier, E.; Kääb, A.; Bolch, T. and Hoelzle, M. Multi-decadal mass balance series of three Kyrgyz glaciers inferred from modelling constrained with repeated snow line observations, The Cryosphere, 2018,12, 1899-1919.

Pelto, M.: Utility of late summer transient snowline migration rate on Taku Glacier, Alaska, The Cryosphere, 2011, 5, 1127-1133.

Shea, J. M., Menounos, B.; Moore, R. D. and Tennant, C. An approach to derive regional snow lines and glacier mass change from MODIS imagery, western North America. The Cryosphere, 2013, 7, 667-680.

**Response**: We appreciate the Reviewer for the recommendation of the three important references. These references are added in the revised manuscript as follows.

"*Numerous studies (Braithwaite, 1984; Barandun et al., 2018; Rabatel et al., 2005, 2008, 2012; Shea et al., 2013; WGMS, 1991-2013; Xie et al., 1996) have shown that glacier annual mass balance is highly correlated with the ELA or SLA-EMS, and the SLA-EMS enables reconstruction of annual mass balance time series.*"

"*Most previous studies (Barandun et al., 2018; Kundu and Chakraborty,2015; McFadden et al., 2011; Pandey et al., 2013; Rabatel et al., 2005, 2012; Tawde et al., 2016; Zhang and Kang, 2017) of SLA-EMS have focused on local areas and using visual interpretation of Landsat MSS/TM /ETM+ or other high resolution remote sensing images observed near the end of summer.*"

"*In the aspect of using MODIS data to estimate SLA-EMS, Pelto (2011) found that the late summer transient snowline identified from MODIS and Landsat images can provides a means for*

*efficient mass blance assessment on the Taku Glacier, and the transient snowline position identified is not significantly different in MODIS and Landsat images. Later, **Shea et al. (2013)** developed a method to calculate regional transient snowline altitude and SLA-EMS from daily MODIS imagery (MOD02QKM) on glaciers in western North America; a cluster analysis of the cloud-masked visible and near-infrared bands is used to delineate the transient snowline, and the maximum value on the regression fitted curve of transient snowline altitudes in ablation season is used to represent the SLA-EMS (as ELA proxy). However, the method of Shea et al. (2013) is also unsuitable to provide spatial and temporal continuous monitoring of SLA-EMS over a large-scale area (such as the whole of the HMA), because for each glacierized region and ablation season, it requires a lot of repetitive work in selecting images with fewer clouds, delineating and calculating transient snowline altitudes at daily timescales, and extracting the SLA-EMS from transient snowline altitudes. Thus, it is worth studying to develop more efficient method for estimating spatial and temporal continuous SLA-EMS over a large-scale area, using the MODIS snow cover products.*"

The method proposed in this manuscript is different from the Pelto (2011) and Shea et al. (2013), though MODIS images are also used in their studies. The differences are as follows: (1) in Pelto (2011) and Shea et al. (2013), the used data is visible and near-infrared bands of MODIS imagery, which greatly affect by cloud cover. In this manuscript, the daily MODIS fractional snow cover (FSC) products is used, and the cloud cover in the MODIS FSC datasets is effectively removed by the developed cubic spline interpolation cloud removal method. (2) in Pelto (2011) and Shea et al. (2013), for each glacierized region and ablation season, it requires a lot of repetitive work in selecting images with fewer clouds, delineating and calculating transient snowline altitudes at daily timescales, and extracting the SLA-EMS(maximum value) from on the regression fitted curve of transient snowline altitudes (the Fig.4 in Shea et al.). In this manuscript, the MODIS-derived snow covered days (SCD) is adopted to estimate the perennial snow cover, the SLA-EMS can be calculated on a grid-by-grid basis using the area-elevation distribution curve and the perennial snow cover area. Thus, spatial and temporal patterns of SLA-EMS over the whole HMA during 2001-2016 were estimated by this automatic method.

**2)** The footprint of MODIS in steep terrain leads to significant errors in establishing a SLA. This has to be explored, particularly given a grid scale measurement is used, and is not validated against more specific glacier by glacier observation for a few sample grids.

**129:** Given the 500 m pixel size and the average slopes how much accuracy is there for SLA? This is a key issue given you are reporting a grid cell average. This should be validated for a few particular grid cells with Landsat, SPOT or Sentinel observation of SLA in that same grid cell on glaciers. This is done for SCA using Landsat that is a different measure.

**Response to the 2) and 129**: We agree with the Reviewer. In our manuscript, it is necessary to validate the accuracy of the grid scale measurement of SLA-EMS. We have selected 5 grids to evaluate the MODIS-derived SLA-EMS, using Landsat images (TM, ETM+ and OLI) selected from melt seasons. And the discussion for the possible uncertainty and error sources of the method are added in the Section 5. (Discussion).

[Figure]

**Figure 2.** *…The 5 blue grids (G1-G5) indicate the site for SLA-EMS evaluation.*

**Table 2.** *Information about Landsat TM/ETM+/OLI images used in validation of MODIS-derived SLA-EMS.*

| Year | G1 and G2 Path 147, Row 31 | G3 Path 151, Row 33 | G4 Path 150, Row 34 | G5 Path 145, Row 35 |
|---|---|---|---|---|
| 2001 | Aug 13 | Jul 26 | Jul 3, Aug 20, Sep 5 | Aug 25, Sep 2 |
| 2002 | Jul 1, Jul 17, Aug 18 | Aug 30 | Aug 7, Aug 23 | Jul 3 |
| 2003 | Jul 20 | Jul 16, Sep 2 | | Jul 22 |
| 2004 | Jul 22, Aug 7, Aug 23 | Jul 2, Aug 3, Sep 4 | Jul 11, Jul 27, Aug 12 | Jul 8, Aug 25, Sep 10 |
| 2005 | Jul 25, Aug 10, Aug 26 | Jul 21, Aug 6, Sep 7 | Jul 14, Aug 15, Aug 31 | Jul 11, Aug 12, Sep 13 |
| 2006 | Jul 28, Aug 13, Aug 29 | Jul 24, Aug 25, Sep 10 | Jul 17, Aug 2, Sep 3 | Jul 30, Aug 15, Sep 31 |
| 2007 | Aug 8, Aug 24, Sep 1 | Jul 27, Aug 12, Aug 28 | Jul 4, Jul 20, Aug 5 | Aug 10, Aug 18, Sep 3 |
| 2008 | Aug 2, Aug 10, Sep 3 | Jul 29, Aug 14, Aug 30 | Jul 6, Aug 7, Aug 23 | Jul 3, Aug 4, Aug 5 |
| 2009 | Jul 20, Aug 10, Aug 21 | Jul 16, Aug 1, Aug 17 | Jul 9, Jul 25, Aug 2 | Jul 6, Jul 15, Aug 31 |
| 2010 | Jul 15, Jul 23, Aug 16 | Jul 19, Aug 20, Sep 5 | Jul 12, Aug 21, Aug 29 | Jul 9, Jul 17, Aug 26 |
| 2011 | Jul 26, Aug 3, Aug 19 | Jul 22, Aug 23, Sep 8 | Jul 7, Aug 8, Aug 24 | Jul 28, Aug 5, Aug 21 |
| 2012 | Jul 28, Aug 29 | Jul 9, Aug 25, Sep 10 | Jul 17, Aug 2, Aug 18 | Aug 31 |
| 2013 | Jul 23, Jul 31, Sep 1 | Jul 19, Jul 27, Aug 28 | Jul 4, Jul 28, Sep 6 | Jul 17, Aug 2, Aug 10 |
| 2014 | Jul 10, Jul 26, Aug 11 | Jul 14, Aug 7, Aug 31 | Jul 15, Jul 23, Aug 1 | Jul 28, Aug 13, Aug 21 |
| 2015 | Aug 6, Aug 14, Aug 30 | Jul 17, Aug 18, Sep 3 | Aug 19, Aug 27, Sep 4 | Jul 23, Aug 8, Sep 9 |
| 2016 | Aug 8, Aug 24, Sep 1 | Jul 27, Aug 4, Aug 28 | Jul 20, Jul 28, Aug 13 | Jul 2, Jul 17, Jul 25 |
| Total | 43 | 43 | 44 | 41 |

In the secsion    4. Results

"*4.1 Evaluation of MODIS-derived grid (30km) SLA-EMS*

   *To evaluate our method, we compared SLA-EMS of 5 grids manually digitized from*

*high-resolution Landsat images (Table 2) with the automatic measured results from MODIS, during 2001-2016. To be consistent with the DEM data sources of the MODIS-derived SLA-EMS, the 90m SRTM DEM is also used to calculate the SLA-EMS derived from the manual delineation. For each grid and year, the highest snowline is manually digitized as the "truth-value" of SLA-EMS by combining several Landsat images of melt season. The mean absolute error, root mean square error and correlation coefficient are employed to evaluate the reliability of the MODIS-derived SLA-EMS (Table 3). In the 5 validation grids, the mean absolute error of MODIS-derived SLA-EMS compared with the manually-derived (Landsat) values is between 44.9 and 124.7 m, and the RMSE is between 52.3 and 133.4m. Despite these differences between the MODIS-derived SLA-EMS and that from manually-derived (Landsat), the correlation coefficients between them are high (between 0.63 and 0.87), and they are all significant at the 0.01 level. The significant correlations indicate that the proposed method can be used to accurately monitor the interannual variations of SLA-EMS. We believe that the MODIS-derived SLA-EMS with such accuracy in the 30km grids can be applied to investigating the spatiotemporal patterns of SLA-EMS in the HMA."*

**Table 3.** *Comparison of SLA-EMS derived from manual delineation of snowline (Landsat images) and automatic calculation from MODIS, during 2001 to 2016.*

| Site | Mean absolute error (m) | Root mean square error (m) | Correlation coefficient |
|------|------------------------|---------------------------|------------------------|
| G1 | 76.4 | 84.4 | 0.70** |
| G2 | 124.7 | 133.4 | 0.65** |
| G3 | 67.1 | 78.0 | 0.87** |
| G4 | 50.6 | 62.2 | 0.76** |
| G5 | 44.9 | 52.3 | 0.64** |

*\*\* indicate statistical significance at the 0.01 level.*

In the section 5. Discussion

 …

 *"Due to the coarse resolution (500m) of the MODIS data, the proposed method for SLA-EMS monitoring is limited to large scale areas, and it is also why the glacier grids are divided as big as 30km in this study. The uncertainty of the MODIS-derived SLA-EMS may come from different sources of errors: (1) errors occurred due to the pixel size of the remote sensing images, slope and aspect of the terrain, the accuracy of the georeferencing and the quality of the DEM (Rabatel et al., 2002, 2005, 2012); (2) the errors in MODIS snow mapping algorithm (Hall and Riggs, 2007; Rittger et al., 2013) and cloud removal method (Tang et al., 2013), although the MODIS SCD threshold is calibrated in this method."*

**3)** The interpretation of the SLA variation being dependent on latitude and regional mean elevation does not capture key drivers of this including differences in moisture sources, and seasonal distribution of precipitation.

**288:** How much of this is latitude versus level of maritime climate influence, or degree to which the glacier is a summer accumulation type? Many references have examined this issue.

**397:** The stipulation that elevation and latitude are the key variables is not supported by much of the literature that indicates how impacted by the summer monsoon and the winter westerlies is a key variable depending on location. That they are two well correlated parameters is accurate.

**Response to the 3), 288 and 397**: As you know, the spatial pattern (or spatial difference) of SLA-EMS is integratedly influenced by climatic factors, such as precipitation, temperature, solar radiation, air humidity and so on. And the climate in HMA is mainly affected by latitude, atmospheric circulation and topography. Thus, we can say, the latitude, atmospheric circulation, and topography are the fundamental factors that affect the SLA-EMS over the HMA. The factors as you pointed out (the moisture sources resulted by summer monsoon and winter westerlies, and the precipitation) are all fundamentally affected by the latitude, atmospheric circulation, and topography in the HMA.

In this paper, we mainly focus on the spatial and temporal patterns of the SLA-EMS in the whole of the HMA, and the possible cause for the inter-annual changes of SLA-EMS from the perspective of temperature and precipitation. Thus, in Section 4.1 Spatial Pattern of SLA-EMS, we just do a simple analysis of the indicated spatial patterns between the SLA-EMS and latitude and topographic elevation from the Fig. 8, 9 and 10. And the relationship between them is obvious.

The influences of different climatic factors on the spatial pattern of SLA-EMS are complicated. For instance, there is no obvious relationship between the spatial distribution of SLA-EMS and precipitation (list in follow figure). **Quantifying the effects of various factors on the spatial pattern of SLA-EMS is an extraordinary challenge for our further studies**.

[Figure]

**Based on your suggestions, we have made the following changes.**
In the section   5. Discussion
…
(add to the Line 408) "*In addition, the HMA is mainly influenced by three atmospheric circulations: the Indian and East Asian summer monsoon, and the westerlies. The moisture transported by these atmospheric circulations (such as the summer southwest monsoon from Indian Ocean) is difficult to reach the interior high-elevation regions of HMA, due to huge shielding of the topography. Thus, the moisture differences controlled by topography may be another reason for the spatial patterns between the SLA-EMS and topographic elevation.*"
In the section 4.3. Correlations between SLA-EMS, Temperature and Precipitation. We

have changed the temperature and precipitation data of meteorological stations into ERA5 reanalysis data, because the meteorological stations are scarce in glaciated regions. This time, the correlation coefficients between the SLA-EMS, temperature, and precipitation during 2001 to 2016 are calculated on a grid-by-grid basis, as shown in **Fig. 13**. And the result analysis in this manuscript will be modified accordingly.

[Figure]

*Figure 13. The correlation coefficients between the SLA-EMS, temperature, and precipitation during 2001 to 2016.*

*Table 6. The averages of Pearson correlation coefficients between the SLA-EMS, temperature, and precipitation for different subregions in the period of 2001-2016.*

| Regions | Summer temperature | Annual temperature | Summer precipitation | Annual precipitation |
|---|---|---|---|---|
| S and E Tibet | 0.55 * | 0.38 | -0.05 | -0.09 |
| Hengduan Shan | 0.13 | 0.01 | 0.06 | 0.01 |
| Qilian Shan | 0.75 ** | 0.61 * | 0.26 | 0.17 |
| Inner Tibet | 0.52 * | 0.36 | -0.08 | -0.19 |
| E Tien Shan | 0.58 * | 0.17 | -0.18 | -0.16 |
| W Tien Shan | 0.55 * | 0.11 | 0.05 | -0.13 |
| E Himalaya | 0.46 | 0.23 | 0.03 | 0.01 |
| E Kun Lun | 0.64 ** | 0.48 | 0.15 | 0.14 |
| C Himalaya | 0.37 | 0.36 | 0.13 | -0.16 |
| Pamir | 0.77 ** | 0.41 | 0.04 | -0.53 * |
| W Himalaya | 0.64 ** | 0.52 * | -0.02 | -0.40 |

| | | | | |
|---|---|---|---|---|
| *Altay and Sayan* | *0.67 \*\** | *0.30* | *-0.07* | *-0.32* |
| *Hissar Alay* | *0.67 \*\** | *0.13* | *0.15* | *-0.61 \** |
| *Hindu Kush* | *0.86 \*\** | *0.46* | *-0.36* | *-0.68 \*\** |
| *W KunLun* | *0.63 \*\** | *0.22* | *-0.05* | *-0.08* |
| *Karakoram* | *0.57 \** | *0.32* | *0.19* | *-0.28* |

**10:** Why use a new acronym instead of the excepted terminology of transient snow line (TSL) for observations of the snow line not at the end of the melt season or snow line altitude (SLA) if it is the end of the melt season and is equivalent to the ELA, than just use ELA.

**Response:** In this paper, we used the snowline altitude at the end of melting season (SLA-EMS) throughout manuscript. It is not a new concept, for example, the same "snowline at end of melting season" in Pandey et al. (2013); and it is often called as the snowline at the end of the ablation season (Meier and Post, 1962), or the end-of-summer snow line altitude (Clare et al., 2002), or the snowline altitude at the end of the hydrological year (Rabatel et al., 2012), and so on.

   Equilibrium line is the boundary between the accumulation and ablation zone of the glacier. It is a theoretical line which is irregularly distributed on the glacier. Although numerous studies have shown that the SLA-EMS approximates can be considered as approximation (or representative) of the ELA, we think it is not rigorous to change the "SLA-EMS" as "ELA" in this paper.

Pandey, P., Kulkarni, A. V., and Venkataraman, G.: Remote sensing study of snowline altitude at the end of melting season, Chandra-Bhaga basin, Himachal Pradesh, 1980–2007. Geocarto Int., 28, 311-322, 2013.

Meier, M.F. and Post, A.S., 1962. Recent variations in mass net budgets of glaciers in western North America. International Association of Scientific Hydrology Publication, 58 (Symposium at Obergurgl 1962 - Variations of Glaciers), 63-77.

Clare, G.R., et al., 2002. Interannual variation in end-of-summer snowlines of the southern alps of New Zealand, and relationships with southern hemisphere atmospheric circulation and sea surface temperature patterns. International Journal of Climatology, 22, 107–120.

Rabatel, A., Bermejo, A., Loarte, E., Soruco, A., Gomez, J., Leonardini, G., Vincent, C 562 ., and Sicart, J. E.: Canthe snowline be used as an indicator of the equilibrium line and mass balance for glaciers in the outer tropics? J. Glaciol., 58, 1027-1036, 2012.

**45:** Check, Flint (1971) not a good interpretation of snowline.

**Response:** We have changed "*…the snowline defines the lowest altitude of the perennial snow cover (Flint, 1971), …*" into "*…the snowline is the boundary separating perennial snow cover from seasonal snow cover areas(Huang et al., 2006; Wu et al.,2008)…*".
*Wu, G., Wang, N., Hu, S., Tian, L., Zhang, J.: Physical geography. Higher education press, Bejing, China, 241-242, 2008.*
*Huang, Z., Zhang L.: Dictionary of earth sciences. Geology press, Bejing, China, 351-352, 2006.*

**56:** There are useful references here including from the HMA that illustrate more frequent SLA observation

(Das and Chakraborty, 2015)

Das, S, and Chakraborty, M: Delineation of glacial zones of Gangotri and other glaciers of Central Himalaya using RISAT-1 C-band dual-pol SAR. International Journal of Remote Sensing 36(6):1529-1550.

**Response:** This reference is added in the revised manuscript.

 "*Most previous studies (Barandun et al., 2018;Kundu and Chakraborty,2015; McFadden et al., 2011; Pandey et al., 2013; Rabatel et al., 2005, 2012; Tawde et al., 2016; Zhang and Kang, 2017) of SLA-EMS have focused on local areas and using visual interpretation of Landsat MSS/TM /ETM+ or other high resolution remote sensing images observed near the end of summer*.*"*

*Kundu, S., Chakraborty, M.: Delineation of glacial zones of Gangotri and other glaciers of Central Himalaya using RISAT-1 C-band dual-pol SAR. Int. J. Remote Sens., 36, 1529-1550, 2015.*

**79:** This is not true, note work of Barundun et al (2018): "The integration of TSL observations into conventional modelling is shown to be highly beneficial for filling the gaps in long-term SMB series for periods for which direct glaciological measurements were discontinued or are missing completely."

**369:** I do not see how this study is a precedent for using MODIS for snow cover mapping regardless of region or end product.

**Response to 79 and 369:** The Reviewer may not catch my point in Line79 and Line369.

   We consider the main contribution of this manuscript is the spatial detailed (on a grid-by-grid basis) monitoring of the changes in SLA-EMS (2001-2016) over the whole of the HMA, base on the advantage (the high time temporal resolution) of MODIS fractional snow cover (FSC) product and the developed cloud removal method. However, previous studies only focused on individual or several glaciers, or on a specific small area.

   We also think the developed MODIS-derived SLA-EMS dataset and the related spatiotemporal changes results for the whole of the HMA can be considered as a practice without precedent, despite the coarse resolution (30km) of the dataset.     Other comment can see the **Response to 1).**

   To make it more clearly, we have modified the L79 and L369 slightly:

To L79:    "*However, they are difficult to assess SLA-EMS changes in a continuous time and space for a large-scale area (such as the whole of the HMA), due to the 16-day or longer revisit period (including cloud cover) and relatively small swath width of Landsat and other high resolution remote sensing images.*"

To L369:    "*This study can be considered as a precedent toward using MODIS snow cover products to assess SLA-EMS at the whole of the HMA for better understanding of water resources and climatology of cold region.*"

**84:** Shea et al (2013) used MODIS for regional snowline altitude assessment just as you propose. There basic approach is "We describe a method to calculate regional snow line elevations and annual equilibrium line altitudes (ELAs) from daily MODIS imagery (MOD02QKM) on large glaciers and icefields in western North America. An automated cluster analysis of the cloud-masked visible and near-infrared bands at 250m resolution is used to delineate glacier facies

(snow and ice) for ten glacierized regions between 2000–2011. For each region and season, the maximum observed value of the 20th percentile of snowcovered pixels is used to define a regional ELA proxy."

**87:** Pelto et al (2011) compared MODIS and Landsat for snow line identification: "The MODIS imagery is from band 1 which has a resolution of 250 m. With the average surface slope of 1.6â°U ¿e this yields an error of less than ±10 m in elevation for TSL. A comparison of a Landsat image and MODIS image from 29 July 2009 is provided (Fig.2). It is evident that though some detail is lost the TSL position identified overall is not significantly different."

**Response to 84 and 87:**   The important references (Shea et al., 2013; Pelto, 2011) are added in the revised manuscript. See the **Response to 1).**

In addition, in line 87, the **Evaluation studies** in line 87 are the evaluation for the accuracy of snow mapping in MODIS snow cover products, not the evaluation of the snow line identification.

*L87 : "Evaluation studies have suggested a high accuracy of MODIS snow cover products under clear skies, when comparing with the in-situ observations and other higher resolution satellite data at both regional and global scales (Hall and Riggs, 2007; Klein et al., 2003; Tang et al., 2013)"*

**142:** How many report ELA observations?

**218:** Given that many of these glaciers report ELA to WGMS, which provides a more direct measure of the adequacy of your method, it would be appropriate to provide this measure. The mass balance provides a correlation that is similar, however, the standard deviation between the methods is meaningless with the different units. This could be done collectively versus glacier by glacier.

**Response to 142 and 218:** In the Fluctuations of Glaciers Database (2017) (WGMS), the ELA observations are mainly calculated from a lot of mass balance observations (fitting the contour line of annual mass balance, and calculating the altitude at the zero annual mass balance). Thus, the ELA observations may be less than the annual material balance observations. And in the calculating, it may also bring uncertainty to the ELA observations. The numbers of the two kinds of observations are compared, as shown in the following table. The numbers of ELA observations are less than that of annual mass balance. In addition, previous studies have been more frequent to comparing the relationship between SLA and annual mass balance, rather than that of SLA and ELA. Therefore, we selected the observations of annual mass balance in this research.

Number of observations for the annual mass balances (AMB) and ELA in the 12 measured glaciers of the 16 years.

| Glaciers | Number of AMB Observations | Number of ELA Observations |
|---|---|---|
| XIAO DONGKZMADI | 10 | 0 |
| CHHOTA SHIGRI | 12 | 12 |
| MALIY AKTRU | 12 | 12 |
| LEVIY AKTRU | 12 | 12 |
| MERA | 8 | 8 |
| VODOPADNIY (NO.125) | 12 | 11 |
| TS.TUYUKSUYSKIY | 16 | 16 |

| | | |
|---|---|---|
| URUMQI GLACIER NO. 1 | 16 | 16 |
| QIYI | 7 | 0 |
| CHORABARI | 7 | 0 |
| PARLUNG NO. 94 | 10 | 8 |
| POKALDE | 6 | 5 |

**189:** "However, the snow area with MODIS SCD≥365d fails to really indentify the perennial snow area, due to the affect of the annual cumulated errors in MODIS snow mapping algorithm and cloud removal method."

**Response:** We have revised it as you suggested. Thanks.

**201:** Why is the 25km2 glacier area of the 30km2 grid chosen?

**Response:** Please note that the area of the glacier grid (cell size 30km) is 900 km$^2$. If glacier area in the grid is too small, its perennial snow cover would be difficult to accurately identify by MODIS, due to the low resolution (500m).
"*Due to the low resolution (500m) of MODIS, for each identified glacier grid, the area of glacier cover is ensured to be more than 25km$^2$ based on overlay analysis with the glacier inventory data.*"

**214:** The correlation from 332d to 347d is relatively consistent indicating this is a good window, and 347d alone does not have to be relied upon if imagery is poor. If the this time of 347 days shifts that is a measure too.

**Response:** In order to accurately estimate the perennial snow cover (minimize the annual cumulated errors in MODIS snow mapping algorithm and cloud removal method), the MODIS SCD threshold was calibrated using both the glaciers annual mass balance observations and Landsat images. We really can't say the MODIS SCD threshold of 347d is perfect for any region, but it may be the most suitable threshold from the whole the of the HMA. We agree with the Reviewer that it would be more appropriate to apply shifting thresholds, but it is impossible to calibrate all the thresholds for the 744 grids on a grid-by-grid and year-by-year basis. Just as the MODIS binary snow cover product, the same threshold of NDSI is used globally for snow mapping.

**241:** In figure 6 it is provide legend for the various regions. The continued declining ratio of Landsat/MODIS beyond 347d suggests that the melt season is continuing and the SLA would still be rising.

**Response:** This is not true. For any region and year, the MODIS SCD is constant. In figure 6, with the SCD threshold changing from 280 to 365, the MODIS-derived snow-covered area (with SCD ≥ threshold) must be decreasing gradually. Therefore, it cannot indicate the melt season or the seasonal variation of snow cover.

**258:** This method appears to be quite useful for providing a comparable SLA elevation across the region annually, even if it is not overly accurate to a glacier in particular grid cells or for mass balance assessment.

**Response:** Yes. This method can effectively reflect the annual fluctuation of SLA-EMS on a grid-by-grid (30km x 30km) basis. In theory, it can also reflect the fluctuation of the annual mass balance of glaciers on the 30km grids scale. However, the quantitative relationships between SLA-EMS and glacier annual mass balance at the scale of 30km grids needs to be further studied in the future work using adequate spatially resolved glacier annual mass balance data, for example using time series of DEMs derived from ASTER optical satellite stereo-images.

**287:** How does fit with the results of Barundun et al (2018) from the Tien Shan and Pamir-Altay?

**Response:** We have examined the transient snow line (TSL) data (download from: https://doi.org/10.5194/tc-12-1899-2018-supplement) on the three glaciers of Barundun et al (2018). The following table shows the comparison of the highest altitude of TSL in the three glaciers from Barundun et al (2018) and the MODIS-derived SLA-EMS in the corresponding grids (30km) in this research. However, we believe that the results of Barundun et al (2018) cannot be used as the "truth value" for the accuracy evaluation of the MODIS-derived SLA-EMS in this paper. There are two main reasons. 1) The area of a single glacier is too small to be used for the comparison of SLA-EMS of the 30km glacier grid (900 $km^2$) in this paper; 2) It is doubtful that almost all the highest altitudes of TSL (in the 16 years) on the three glaciers occurred at the end of September in the results of Barundun et al (2018); This is inconsistent with the research of Tang et al (2017), in which the months with smallest snow cover are in July to August (the Figure 4 and 5).

Tang, Z., Wang, X., Wang, J., Wang, X., Li, H., and Jiang, Z.: Spatiotemporal variation of snow cover in Tianshan mountains, Central Asia, based on cloud-free MODIS fractional snow cover product, 2001–2015. Remote Sensing, 9, 1045, 2017.

Abramov Glacier (39°36.78' N, 71°33.32' E), with the extent of about 24 $km^2$.
Golubin Glacier (42°26.94' N, 74°30.10' E), 5 $km^2$.
Glacier no. 354(41°47.62' N, 78°9.69' E), 6.4 $km^2$.

Table.   The comparison of the highest altitude of TSL in the three glaciers from Barundun et al (2018) and the MODIS-derived SLA-EMS in the corresponding grids (30km),

| Glacier | Years | Mean absolute error (m) | Root mean square error (m) | Correlation coefficient |
|---------|-------|-------------------------|----------------------------|-------------------------|
| Abramov | 2001-2016 | 90.69 | 105.58 | 0.75** |
| Golubin | 2001-2016 | 302.56 | 321.64 | 0.48 |
| Glacier No.354 | 2004-2016 | 91.77 | 106.27 | 0.63* |

** and * indicate statistical significance at the 0.01 and 0.05 level, respectively.

**291:** Both altitude and elevation are used in this sentence, are you referring to the average elevation of the grid cell?

**Response:** Yes, it is the average elevation of the grid. We have changed the "altitude" as "elevation".

**306-307:** Your percentages refer to discrete linear responses and should be grouped in a way that the 100% is reached in a clear way. Lump together the significance groupings. Lump together the slope groupings.

**Response:** We have added two Tables as follows.

*Table 4. The percentages of the number of grids for different change trend groupings of the SLA-EMS (calculated from Fig. 11a).*

| Trend | Change trend (m yr$^{-1}$) | | | | | | |
|---|---|---|---|---|---|---|---|
| | <-8 | -8~-3 | -3~0 | 0~3 | 3~8 | 8~15 | >15 |
| percentage(%) | 1.47 | 4.41 | 12.21 | 38.82 | 32.06 | 8.82 | 2.21 |

*Table 5. The percentages of the number of grids for different significance groupings of the change in SLA-EMS (calculated from Fig. 11b).*

| Significance | Significant increase | Nonsignificant increase | Significant decrease | Nonsignificant decrease |
|---|---|---|---|---|
| percentage(%) | 26.47% | 55.44% | 0.88% | 17.21% |

**311-313:** Quantify what % of decreasing trends are from the indicated regions.

**Response:** we have changed it.

*"The grids with significant increased SLA-EMS are mainly distributed in east Tien Shan (58.33%), west Tien Shan (54.41%), east Himalayas (84.85%), central Himalayas (30%), inner Tibet (43.21%), and the south and east Tibet (70%)."*

**347:** The comment about the rising SLA and water resources is a generic statement that needs support or removal. Thayyen and Gergan (2010) have described how the runoff from summer accumulation type glaciers is less of a resource than for other areas. If the melt season expands into the fall months as has been noted, this is a lower flow period and water resources could be increased with more glacier melt.

**Response:** Here, we only commented that the rising SLA-EMA may result in significant changes in water resources. But not specifying how it changes (decrease or increase).

**390:** That SLA is a good indicator of mass balance is well established. In this case an indicator is not a substitute for any of the other methods that provide an actual quantity that can be validated with one of the other methods.

**Response:** Here, we have deleted "*The MODIS extracted grid SLA-EMS can be a good indicator of large-scale glaciers annual mass balanc*e".

**420:** They do indicate declining mass balance.

**Response:** Thanks. We have deleted the word of "may".

**467:** The rising snowlines have already led to a decline in mass balance and mass flux down glacier. This is a continuation of regional mass loss that has driven thinning and a slowdown in glacier movement in 9 of 11 regions in HMA from 2000-2017 (Dehecq et al 2019).

Dehecq, A., N. Gorumelon, A. Gardner, F. Brun, D. Goldberg, P. Nienow, E. Berthier, C. Vincent, P. Wagnon, and E. Trouve, 2019: Twenty-first century glacier slowdown driven by mass loss in High Mountain Asia. Nature Geoscience 12, 22–27.

**Response:** The changing in SLA-EMS reflects the changes in the annual (net) mass balance (the negative correlation). It is precisely because the most of glaciers over the HMA have been losing mass in the last 16 years (i.e., the average annual mass balance has been negative), the further rising of the SLA-EMS will accelerate the negative annual mass balance.

Changes in L467: "*Under the background of the generally losing glaciers mass over the HMA, if the global warming continues, the rising of the SLA-EMS may accelerate the negative mass balances of the most glaciers, lead to change the flow regimes and water availability, thus impacting ecosystem, agriculture and water resources in the densely populated downstream areas.*"

And the reference (Dehecq et al 2019) are added in the section 5. Discussion (L421):

"*…Under the background of the generally losing glaciers mass in these areas (Brun et al., 2017; Dehecq et al., 2019) (i.e. average annual mass balance is already negative), if the SLA-EMS continues to rise as a result of global warming, it will accelerate the negative mass balances of the glaciers.*"